# INTERMEDIATE LAYERS CAN BE SELF-HARD NEGATIVE GENERATOR FOR LARGE LANGUAGE MODEL BASED RECOMMENDATION

## ABSTRACT

Large language models (LLMs) have gained significant attention for their usage in recommender systems. One typical method to adapt LLMs for recommendation is Supervised Fine-tuning (SFT), and subsequent studies introduce preference learning to incorporate negative samples into the training process. However, the negative samples used in existing preference learning methods are sampled at the sequence-level in an offline process, making them less discriminative and informative when adapting LLMs to recommendation tasks with large negative item spaces. To address these challenges, we propose **ILRec**, a novel preference fine-tuning framework for LLM-based recommender systems, which utilizes self-hard negative signals extracted from intermediate layers to enhance preference learning for LLMs. Specifically, we first extract self-hard negative tokens from intermediate layers, which serve as fine-grained negative signals and dynamically reflect the model's preference learning process. To incorporate these negative signals into training, we devise a fine-tuning framework consisting of two components: cross-layer preference optimization and cross-layer preference distillation, which enables the model to effectively distinguish the negative signals and enhance the informativeness of negatives generated by intermediate layers. Additionally, we introduce a small collaborative filtering model to assign reward to each penalized token, preventing potential over-penalization of false negatives. Extensive experiments on three datasets demonstrate ILRec's effectiveness in enhancing the performance of LLM-based recommender systems. The source code is available at https://anonymous.4open.science/r/ILRec-6FFE.

## 1 INTRODUCTION

Sequential recommendation aims to predict the next item a user will likely interact with based on the historical interaction sequence (Kang & McAuley, 2018; Hidasi et al., 2016). Recently, as large language models (LLMs) have shown remarkable capabilities in text understanding and generation (Clusmann et al., 2023), enhancing recommender systems with LLMs has received widespread attention (Bao et al., 2023b; Geng et al., 2022). As one typical paradigm, LLMs are leveraged to directly generating the identifier (*i.e.,* a sequence of tokens) of the item for recommendation (Liao et al., 2024b; Bao et al., 2023a).

To adapt LLMs for recommendation, early work (Zhang et al., 2023; Zheng et al., 2024) adopts Supervised Fine-Tuning (SFT), formatting interacted item sequences as text-based instructions and takes the identifiers of ground-truth items as responses for training. Considering the importance of negative samples in recommendation tasks, some studies (Chen et al., 2024; Liao et al., 2024a; Gao et al., 2024) further utilize DPO-based (Rafailov et al., 2024) preference alignment techniques to incorporate both positive and negative items in training, enabling LLMs to learn user ranking preferences. In particular, they propose to extract negative items via random sampling (Chen et al., 2024) or self-generation (Liao et al., 2024a; Gao et al., 2024), and design specific training objectives like uncertainty-based DPO (Liao et al., 2024a) and self-play learning (Gao et al., 2024) to better leverage information from negative samples.

Despite remarkable progress, existing methods exhibit limitations in both the discriminativeness and informativeness of sampled negatives. As illustrated in Figure 1, these limitations manifest in two

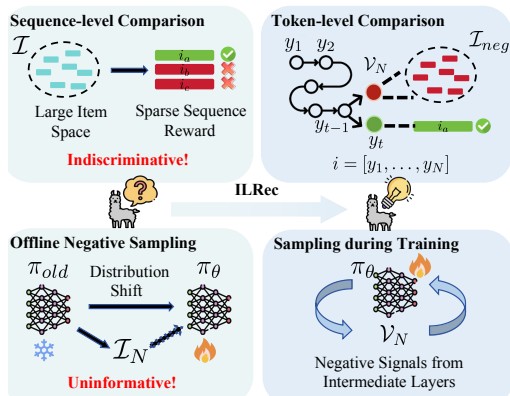 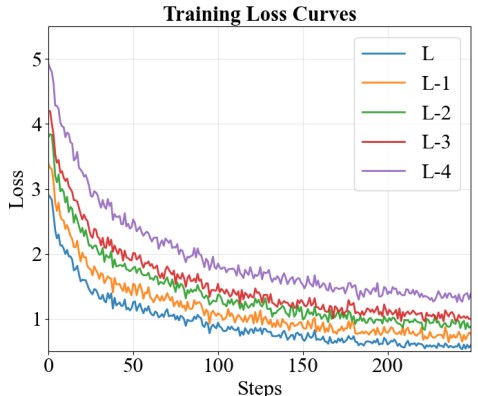

Figure 1: Limitations of traditional negative sampling methods and solutions provided by ILRec, aiming at extracting more fine-grained and informative negative signals.

Figure 2: Training loss curves of different layers in LLaMA3.1-8B on Instrument dataset using LC-Rec. $L$ denotes the final layer, while $L - k$ denotes the $k$-th layer before the final layer.

primary ways: First, current approaches compare ground-truth items against a few negative items and assign rewards to the entire sequence of tokens in responses. Learning from these sparse and coarse-grained rewards makes it challenging for LLMs to capture fine-grained token patterns and user preferences, especially considering large amounts of items as potential candidates in full ranking tasks. Second, the quality of these negative samples is often uninformative to guide the model's optimization. Most existing methods predominantly rely on negative samples collected offline from outdated policy models before further training. As a result, these negative samples struggle to keep pace with the distributional changes of the current updated policy model (Guo et al., 2024; Levine et al., 2020), and do not represent the most informative or worth-learning negatives. This mismatch hinders the model's ability to distinguish hard negatives during training, ultimately leading to suboptimal performance. In addition, the additional preference alignment stage and the increased number of negative samples both introduce extra training and sampling costs, thereby reducing the efficiency of adapting LLMs for recommendation tasks.

To address these limitations, our main idea is to dynamically self-generate and utilize fine-grained negative samples during the training process. Recent studies (Li et al., 2022; Sang et al., 2024) have shown that the outputs of expert models can be optimized by contrasting them with those of non-expert models, since the predictions of non-expert models often contain erroneous patterns or suboptimal choices. This observation provides a valuable perspective for negative sampling strategies. Considering the internal structure of LLMs, the intermediate layers can also be viewed as models with enough predictive capabilities but weaker than the final output layer, as illustrated in Figure 2. This makes them highly promising of dynamically generating appropriately-hard negatives for model optimization during training. Therefore, we propose to utilize the intermediate layers of LLMs as negative generators during training, and extract tokens with high generated probabilities from the intermediate layers' outputs as negative signals. These negative signals offer three primary merits: First, these negatives are extracted in token-level instead of sequence-level. This implicitly extends the negative space and provides LLMs with an accurate and fine-grained comparison between the positive item and various negative items, adapting the model to large candidate-item spaces effectively. Second, since the intermediate layers are jointly optimized during training, the negative signals can dynamically reflect the current preference learning process of the model. They are informative enough to be distinguished and penalized in the final output. Third, the extraction of negative signals from intermediate layers can be seamlessly integrated into SFT within one forward process, which is efficient for adapting LLMs for recommendation.

To this end, we propose **ILRec**, an effective fine-tuning framework for LLM-based recommender systems, which utilizes self-hard negative signals extracted from intermediate layers to enhance preference learning for LLMs. Specifically, we combine intermediate layers with additional prediction layers to get token prediction distributions, from which we select high-probability tokens, excluding the ground-truth token, as self-hard negative signals. This method provides dynamically generated

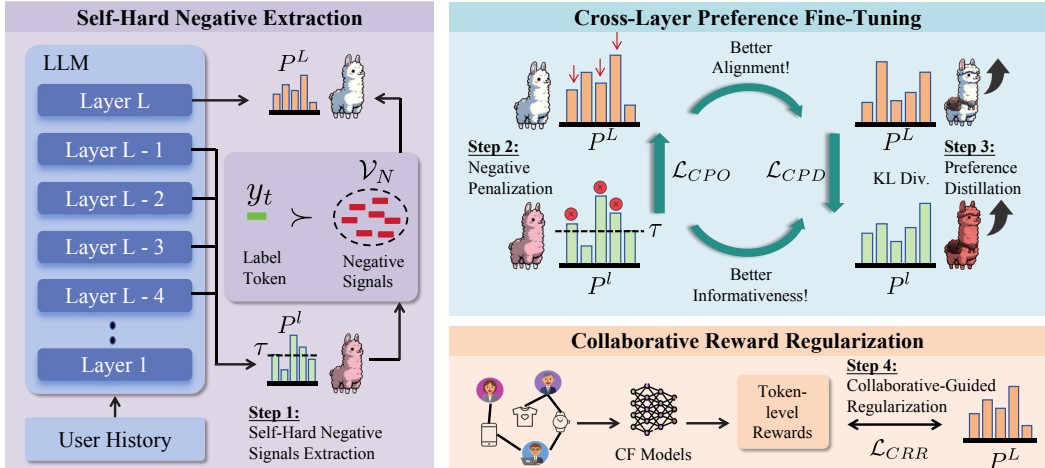

Figure 3: The overall framework of ILRec.

fine-grained negative signals, which enables LLMs to effectively distinguish highly informative negatives from a vast candidate item space. To optimize the generation and utilization of these negative signals, we focus on two points, namely *cross-layer preference optimization* and *cross-layer preference distillation*. Cross-layer preference optimization integrates self-hard negative signals as fine-grained penalty coefficients into the cross-entropy loss, thereby penalizing the corresponding negative tokens in final layer's output. Cross-layer preference distillation employs the output layer as a teacher to supervise the token generation of intermediate layers, which ensures that the self-hard negative signals are reliable and informative enough through training. Furthermore, to address the potential false negatives in extracted tokens, we introduce a small collaborative filtering (CF) model to assign a reward to each penalized token. This helps prevent over-penalization and incorporates CF information during training.

The main contributions of this paper are as follows:

• We present ILRec, a novel LLM-based recommendation framework that extracts fine-grained self-hard negative signals from intermediate layers for preference optimization.

• We propose the cross-layer preference optimization and cross-layer preference distillation for better generalization and utilization of self-hard negative signals during LLM fine-tuning. We also employ collaborative reward regularization to mitigate over-penalization.

• Empirical evaluations on various datasets and scenarios demonstrate the effectiveness of ILRec.

## 2 METHODOLOGY

### 2.1 OVERVIEW OF OUR APPROACH

**Problem Formulation.** Given the item set $\mathcal{I}$, let $S = [i_1, \ldots, i_n]$ denotes the user's historical interactions in chronological order. The goal of traditional sequential recommendation is to predict the next item $i_{n+1}$ for the user based on historical interactions. For LLM-based sequential recommendation, the task is reformulated to an instruction-following paradigm. Given a prompt $X$—containing task descriptions and the sequence of item identifiers in $S$, the LLM is trained to generate the identifier of the next item $Y$ via cross-entropy loss as follows, where $y_{<t}$ denotes tokens before $y_t$ in $Y$:

$$\mathcal{L}_{sft} = -\sum_{t=1}^{|Y|} \log(P(y_t|X, y_{<t})). \tag{1}$$

**Limitations of Negatives used in DPO-based Recommendation Methods.** Large Language Model Based Sequential Recommendation requires LLMs to generate the identifier (*i.e.,* a sequence of tokens) of the item for prediction. In traditional NLP tasks, the response can be highly diverse and

only a few key tokens are critical for conveying the meaning. However, in LLMRec tasks, the token generation process closely resembles a search process in the item space. Each previously generated token plays a crucial role, as it progressively narrows down the possible items and directly influences the final prediction. This makes the generation in LLMRec tasks much more sensitive to training samples. While DPO brings the benefits of leveraging negative samples to LLMRec, there are still certain limitations of applying the extracted negative samples, especially when the negative item space is extremely large (As shown in Figure 1):

- **Indiscriminativeness**: The DPO-based methods align the LLMs using a few of sequence-level negative items, which can be regarded as providing sparse and coarse-grained rewards during training. When facing a large negative item space, these methods make it difficult for LLMs to capture fine-grained token-level patterns and user preferences.

- **Uninformativeness**: Current Approaches sample negatives by random sampling (Chen et al., 2024) or from old policies before optimization. Due to distributional shifts that occur during training, the previously sampled negatives may not be informative or challenging for LLMs to distinguish, leading to suboptimal performances (As shown in Table 1).

**Solution Overview.** To address the challenges posed by large candidate-item spaces and sparse positive feedback in recommendation tasks, we propose a novel framework that leverages fine-grained self-hard negative signals extracted from intermediate layers of LLMs to guide model optimization. Our approach consists of three key components. (1) *Self-Hard Negative Extraction from Intermediate Layers* leverages intermediate layers of the LLM to extract self-hard negative signals. Specifically, high-probability tokens (excluding ground-truth tokens) from these layers are selected as token-level negatives, enabling the model to better align with fine-grained user preferences and effectively handle large negative-item spaces during training. (2) *Cross-Layer Preference Fine-tuning* introduces a framework comprising Cross-layer Preference Optimization (CPO) and Cross-layer Preference Distillation (CPD). CPO penalizes negative signals in the output logits by adding a penalty coefficient to the cross-entropy loss, while CPD improves the quality of negative signals by distilling knowledge from the final output layer to intermediate layers. Jointly training with both modules enhances the model's ability to learn from negative signals. (3) *Collaborative Reward Regularization* employs a lightweight collaborative filtering (CF) model to assign token-level rewards to penalized tokens, preventing excessive penalization and incorporates CF information into the training process. The overall framework of the proposed approach is shown in Figure 3.

## 2.2 SELF-HARD NEGATIVE EXTRACTION FROM INTERMEDIATE LAYERS

To resolve the indiscriminative and uninformative problem of current negative sampling strategies, we seek to dynamically self-generate and utilize fine-grained negative samples during the training process. Recent studies (Li et al., 2022; Sang et al., 2024) have shown that the outputs of expert models can be optimized by contrasting them with those of non-expert models, since the predictions of non-expert models often contain erroneous patterns or suboptimal choices. This observation provides a valuable perspective for negative sampling strategies. Considering the internal structure of LLMs, the intermediate layers can also be viewed as models with enough predictive capabilities but weaker than the final output layer, as illustrated in Figure 2. This makes them well-suited of dynamically generating appropriately-hard negatives for model optimization during training. Therefore, we propose a new negative extraction method that extracts token-level self-hard negative signals from the intermediate layers. The extraction process primarily consists of two parts: (1) Acquiring Ensemble Logits from Intermediate Layers, and (2) Extracting Self-hard Negative Signals from Ensemble Logits.

**Acquiring Ensemble Logits from Intermediate Layers.** To get the negative signals from intermediate layers, we first need to obtain the logits from each intermediate layer. Given the input token $y_t$ at step $t$, the embedding layer together with the transformer layers will generate the corresponding hidden vector $\boldsymbol{h^l}$ at the $l$-th layer. Then, we apply the additional prediction layer $\phi(\cdot)$ to convert the hidden vector $\boldsymbol{h^l}$ into the logits $P^l$ to formulate the LLM's generated values over the vocabulary $\mathcal{V}$:

$$P^l = \phi(\boldsymbol{h^l}) \in \mathbb{R}^{|\mathcal{V}|}, \tag{2}$$

where $|\mathcal{V}|$ denotes the total size of the vocabulary set $\mathcal{V}$. Subsequently, we propose to ensemble the logits of intermediate layers for the extraction of negative signals. Since there is a significant gap between the shallow layers and the deep layers in LLMs, the information provided by shallow layers are not sufficiently informative for generating challenging negative signals of recommendation tasks. Hence, we select consecutive intermediate layers preceding the final output layer as candidate layers. Specifically, We calculate the average value of each layer's logits to form the ensemble logits $\hat{P}$:

$$\hat{P} = \frac{\sum_{l=L-k-1}^{L-1} P^l}{k}, \tag{3}$$

where $L$ denotes the final output layer and $k$ indicates the number of candidate layers before $L$.

**Extracting Self-hard Negative Signals from Ensemble Logits.** Subsequent to acquiring the ensemble logits from intermediate layers, our purpose is to extract the predicted tokens as fine-grained negative signals from non-expert intermediate layers. Hence, we propose selecting those tokens that have high generated probabilities as self-hard negative signals. We set a threshold $\tau$ for selecting these signals, which is positively correlated with the generated probability of the ground-truth token. In this way, our approach can dynamically select self-hard negative signals according to the accuracy of prediction in the ensemble logits. The calculation of threshold $\tau$ is as follows:

$$\tau = \alpha \hat{p}(y_t), \tag{4}$$

where $y_t$ denotes the ground-truth token at step t and $\hat{p}(y_t)$ denotes the probability for $y_t$ in the ensemble logits $\hat{P}$. $\alpha$ is the hyperparameter that controls the threshold for selecting tokens. Then, the set of self-hard negative signals $\mathcal{V}_N$ at each step is as follows:

$$\mathcal{V}_N = \{v | v \neq y_t, v \in \mathcal{V}, \hat{p}(v) \geq \tau\}. \tag{5}$$

Compared to DPO-based methods, which leverage a limited number of sequence-level negative items for preference optimization, the extraction of token-level negative signals in ILRec offers three key advantages. First, our negative signals involve the large candidate token space, which aligns better with the large negative item space in recommendation scenarios. In specific, by learning to distinguish between the ground-truth tokens and these negative signals, the generation probabilities for items prefixed with those negative signals will be reduced to some extent. This enables the LLM to explore the large candidate space more stably and learn the preference paradigm more effectively. Second, these negative signals are dynamically self-sampled during the training process, which are informative enough to provide consistently challenging negative signals for model optimization. Third, our negative signals are generated within the SFT training process and do not require additional training or sampling, serving as a stable and efficient preference alignment method for recommendation.

## 2.3 CROSS-LAYER PREFERENCE FINE-TUNING

After extracting negative signals from intermediate layers, we propose incorporating them within the fine-tuning of LLMs. Since the quality of negative samples are crucial for model optimization, we propose a self-evolving fine-tuning method, simultaneously optimizing the generation and utilization of negative signals. It enables the model to effectively learn from negative signals, while continuously enhancing the informativeness of extracted negative signals. Our proposed framework consists of two components: (1) Cross-Layer Preference Optimization and (2) Cross-Layer Preference Distillation.

**Cross-Layer Preference Optimization.** The core idea of preference optimization in recommendation is to learn the comparison between preferred positive items and less preferred negative items. Therefore, we directly integrate negative signals into the cross-entropy loss for fine-grained preference learning. First, we design a penalizing weight $w_v$ for each token $v$ as follows:

$$w_v = \begin{cases} \frac{\exp(\hat{p}(v))}{\sum_{v_n \in \mathcal{V}_N} \exp(\hat{p}(v_n))} & \text{if } v \in \mathcal{V}_N \\ 0 & \text{if } v \notin \mathcal{V}_N \end{cases}. \tag{6}$$

For those tokens that are not involved in $\mathcal{V}_N$, we set their weight as 0 since they have already been well distinguished by the model.

Then, we reformulate the cross-entropy loss in the original SFT objective. For greater clarity and conciseness, we focus on the training process of single predicted logits instead of the total response. The reformulated cross-entropy loss is as follows:

$$\mathcal{L}_{CPO} = -\log \frac{\exp(p^L(y_t))}{\sum_{v \in \mathcal{V}} \exp(p^L(v)(1 + \beta w_v))}, \tag{7}$$

where $p^L(v)$ denotes the generated probability of token $v$ by the final output layer at step t, while $\beta$ is the hyperparameter that controls the degree of penalization for each token. Based on Equation 6 and the gradient analysis provided in Appendix A.8, the penalizing weight is positively correlated with the value of the corresponding token in $P^L$. This means that challenging negative signals for the model to distinguish from positive samples will be penalized more in the final output logits, helping model effectively optimize its comprehension of user preferences during SFT stage. Additionally, since these negative samples are dynamically self-generated within the model during training, there is no need for external negative samples or repeated iterative learning, thus achieving an efficient self-learning and optimization process.

**Cross-Layer Preference Distillation.** At the beginning of the training process, since there exists a significant capability gap between intermediate layers and the final output layer (Luo & Specia, 2024), the ensemble logits generated by intermediate layers may not provide informative and worth-learning negatives for training the model. To dynamically improve the recommendation ability of intermediate layers, we treat them as student models and the final output layer as the teacher model. Then, we leverage the teacher model to supervise the token probabilities generated by student models via distillation, allowing student models to adapt quickly to tasks. Specifically, we calculate the sum of KL Divergence between each student layer's output distribution and the final layer's output distribution:

$$\mathcal{L}_{CPD} = \sum_{l=L-k-1}^{L-1} KL(g(P^l)||g(P^L)), \tag{8}$$

where $g(\cdot)$ denotes the softmax function that output a probability distribution from logits $P$. By distilling the token patterns of the final output layer $P^L$ to each intermediate layer, these layers can quickly adapt to the recommendation tasks and provide informative negative signals that have to be distinguished by the model's output layer.

The final loss of this module is then the sum of the optimization loss in Equation 7 and the distillation loss in Equation 8:

$$\mathcal{L}_{CPT} = \mathcal{L}_{CPO} + \lambda \mathcal{L}_{CPD}, \tag{9}$$

where $\lambda$ is a hyperparameter that controls the weight of the cross-layer preference distillation loss.

## 2.4 COLLABORATIVE REWARD REGULARIZATION

While our fine-tuning method leverages extracted negative signals, it has potential issues. First, some extracted negatives may be false negatives. Over-penalizing them may distort the true preference distribution. Second, the training process does not incorporate collaborative information, potentially leading to recommendation bias. Therefore, we employ a collaborative filtering (CF) model to assign a reward score to each penalized token and reduce the penalty for those with higher rewards.

Firstly, we denote the probability of the CF model recommending item $i \in \mathcal{I}$ to the user as $R(i)$. Then, the reward for each token $v$ within item $i$ at that step can be formulated as follows:

$$r_v = \frac{\sum_{i \in \mathcal{I}_{\leq v}} R(i)}{\sum_{i \in \mathcal{I}_{<v}} R(i)}, \tag{10}$$

where $\mathcal{I}_{<v}$ denotes the set of items that take tokens before $v$ as the prefix, while $\mathcal{I}_{\leq v}$ denotes the set of items using tokens before $v$ together with $v$ as the prefix. This reward function approximates the reward that LLM can receive by generating a specific token from perspectives of CF models. If a token $v$ has a higher $r_v$, it is more likely a false negative token that has been incorrectly penalized. We utilize these rewards as the soft label to optimize the cross-entropy loss in SFT stage:

$$\mathcal{L}_{CRR} = -\sum_{v \in \mathcal{H}} \frac{\exp(r_v)}{\sum_{v_i \in \mathcal{H}} \exp(r_{v_i})} \log \frac{\exp(p^L(v))}{\sum_{v_i \in \mathcal{V}} \exp(p^L(v_i))}, \tag{11}$$

Table 1: The overall performance comparisons between different baseline methods and ILRec. The best and second-best results are highlighted in bold and underlined font, respectively.

| Methods | Instrument | | | | Art | | | | Game | | | |
|---------|-------|--------|--------|---------|-------|--------|--------|---------|-------|--------|--------|---------|
|         | Hit@5 | Hit@10 | NDCG@5 | NDCG@10 | Hit@5 | Hit@10 | NDCG@5 | NDCG@10 | Hit@5 | Hit@10 | NDCG@5 | NDCG@10 |
| Caser   | 0.0502 | 0.0583 | 0.0287 | 0.0334 | 0.0324 | 0.0524 | 0.0208 | 0.0271 | 0.0217 | 0.0423 | 0.0152 | 0.0179 |
| GRU4Rec | 0.0675 | 0.0773 | 0.0516 | 0.0554 | 0.0652 | 0.0786 | 0.0436 | 0.0577 | 0.0406 | 0.0517 | 0.0289 | 0.0365 |
| SASRec  | 0.0619 | 0.0698 | 0.0474 | 0.0502 | 0.0682 | 0.0845 | 0.0541 | 0.0593 | 0.0422 | 0.0598 | 0.0312 | 0.0396 |
| BIGRec  | 0.0786 | 0.1004 | 0.0742 | 0.0799 | 0.0801 | 0.0979 | 0.0704 | 0.0768 | 0.0502 | 0.0677 | 0.0433 | 0.0481 |
| +RosePO | 0.0772 | 0.0983 | 0.0733 | 0.0786 | 0.0771 | 0.0927 | 0.0668 | 0.0732 | 0.0478 | 0.0655 | 0.0408 | 0.0471 |
| +SDPO   | 0.0793 | 0.1016 | 0.0745 | 0.0806 | 0.0795 | 0.0981 | 0.0693 | 0.0762 | 0.0496 | 0.0665 | 0.0420 | 0.0477 |
| +SPRec  | 0.0801 | 0.1021 | 0.0751 | 0.0808 | 0.0810 | 0.0991 | 0.0722 | 0.0784 | 0.0507 | 0.0683 | 0.0437 | 0.0486 |
| +ILRec  | **0.0844** | **0.1091** | **0.0788** | **0.0856** | **0.0856** | **0.1045** | **0.0764** | **0.0852** | **0.0529** | **0.0709** | **0.0455** | **0.0511** |
| LC-Rec  | 0.0888 | 0.1062 | 0.0776 | 0.0832 | 0.0862 | 0.1045 | 0.0725 | 0.0778 | 0.0674 | 0.0984 | 0.0470 | 0.0561 |
| +RosePO | 0.0861 | 0.1006 | 0.0760 | 0.0807 | 0.0870 | 0.1053 | 0.0731 | 0.0783 | 0.0632 | 0.0927 | 0.0441 | 0.0526 |
| +SDPO   | 0.0894 | 0.1069 | 0.0781 | 0.0836 | 0.0868 | 0.1049 | 0.0727 | 0.0779 | 0.0668 | 0.0975 | 0.0456 | 0.0547 |
| +SPRec  | 0.0888 | 0.1041 | 0.0775 | 0.0825 | 0.0875 | 0.1065 | 0.0730 | 0.0786 | 0.0681 | 0.0996 | 0.0475 | 0.0569 |
| +ILRec  | **0.0966** | **0.1143** | **0.0832** | **0.0889** | **0.0922** | **0.1118** | **0.0757** | **0.0821** | **0.0711** | **0.1075** | **0.0489** | **0.0600** |

where $\mathcal{H} = \mathcal{V}_N \bigcup \{y_t\}$ denotes the set of penalized tokens and the ground-truth token. This loss function can be seen as a K-category cross-entropy loss. The soft labels are calculated by the tokens' rewards and a softmax function. By aggregating the CPT loss in Equation 9 and CRR loss in Equation 11, the final loss of ILRec is as follows:

$$\mathcal{L} = \mathcal{L}_{CPT} + \mu \mathcal{L}_{CRR}, \tag{12}$$

where $\mu$ is the hyperparameter that controls the degree of the soft-label rewarding process.

## 3 EXPERIMENT

### 3.1 EXPERIMENTAL SETUP

**Datasets.** We conducted extensive experiments on three subsets of Amazon Review Data (Ni et al., 2019), *i.e.,* "*Musical Instruments*", "*Arts, Crafts and Sewing*" and "*Video Games*". For data pre-processing, we remove unpopular users and items with less than five interactions through five-core filtering. The detailed statistics of preprocessed datasets are presented in Table 3.

**Baseline Models.** We compare our method with the following baselines, including traditional sequential recommendation models, like Caser (Tang & Wang, 2018), GRU4Rec (Hidasi et al., 2016) and SASRec (Kang & McAuley, 2018); LLM-based recommendation models, like BIGRec (Bao et al., 2023a), LC-Rec (Zheng et al., 2024), SDPO (Chen et al., 2024), RosePO (Liao et al., 2024a) and SPRec (Gao et al., 2024). Details are shown in Appendix A.3.

**Evaluation Settings and Implementation Details.** We employ top-$K$ Hit Ratio (HR) and Normalized Discounted Cumulative Gain (NDCG) to evaluate model performances, with $K$ set to 5 and 10. Following prior studies (Kang & McAuley, 2018), we apply the *leave-one-out* strategy to split training, validation, and test sets. Furthermore, to avoid bias introduced by sampling, we conduct full ranking evaluation over the entire item set. More details can be found in Appendix A.4.

### 3.2 OVERALL PERFORMANCE

Table 1 summarizes the overall performance of ILRec. Compared to traditional recommendation models, LLM-based systems demonstrate consistently superior and stable results across all three datasets, consistent with findings from BIGRec and LC-Rec. This highlights the advantage of leveraging LLMs' abilities of language understanding to capture semantic item features.

Among DPO-based methods, SPRec and SDPO generally outperform RosePO under all-ranking settings, which can be attributed to the increased number of negative samples and the penalization of diverse self-hard samples across iterations, which helps the model better explore large candidate spaces. Compared to SFT-based baselines, SPRec achieves relatively stable gains, showing the effectiveness of integrating the Self-Play Mechanism into LLMRec training.

Our proposed ILRec consistently surpasses all baselines on every metric and dataset, achieving notable improvements over BIGRec and LC-Rec. Unlike prior fine-tuning and post-training methods, ILRec extracts fine-grained self-hard negative signals from intermediate layers and employs cross-

Table 2: Ablation study of our method.

| Methods | BIGRec | | | | LC-Rec | | | |
| | Instrument | | Art | | Instrument | | Art | |
| | Hit@10 | NDCG@10 | Hit@10 | NDCG@10 | Hit@10 | NDCG@10 | Hit@10 | NDCG@10 |
|---|---|---|---|---|---|---|---|---|
| ILRec | **0.1091** | **0.0856** | **0.1045** | **0.0852** | **0.1143** | **0.0889** | **0.1118** | **0.0821** |
| w/o $\mathcal{L}_{CPO}$ | 0.1068 | 0.0813 | 0.0987 | 0.0801 | 0.1111 | 0.0852 | 0.1078 | 0.0794 |
| w/o $\mathcal{L}_{CPD}$ | 0.1051 | 0.0805 | 0.1020 | 0.0805 | 0.1124 | 0.0869 | 0.1092 | 0.0805 |
| w/o $\mathcal{L}_{CRR}$ | 0.1078 | 0.0848 | 0.1029 | 0.0838 | 0.1136 | 0.0875 | 0.1112 | 0.0813 |
| w/o $\mathcal{L}_{CPT}$ | 0.0996 | 0.0795 | 0.0982 | 0.0783 | 0.1067 | 0.0843 | 0.1057 | 0.0780 |
| w/o CNS | 0.1059 | 0.0839 | 0.1015 | 0.0799 | 0.1115 | 0.0860 | 0.1097 | 0.0808 |
| FL + $\mathcal{L}_{CPO}$ | 0.1029 | 0.0814 | 0.0996 | 0.0782 | 0.1097 | 0.0844 | 0.1063 | 0.0784 |
| IL + $\mathcal{L}_{CPO}$ | 0.1048 | 0.0827 | 0.1018 | 0.0801 | 0.1118 | 0.0863 | 0.1081 | 0.0798 |

layer preference fine-tuning techniques. This enables the model to dynamically learn from its own fine-grained errors, leading to significant performance enhancements.

## 3.3 ABLATION STUDY

To assess the contribution of each ILRec module, we conduct ablation studies on the Instrument and Art datasets using BIGRec and LC-Rec training paradigms. Table 2 shows results of five variants: (1) *w/o $\mathcal{L}_{CPO}$* eliminates the Cross-layer Preference Optimization module (Equation 7), resulting in overall performance decline and highlighting its necessity for distinguishing ground-truth from cross-layer negatives. (2) *w/o $\mathcal{L}_{CPD}$* removes the distillation process from the final output layer to intermediates layers (Equation 8), also degrading performance and confirming the benefit of teacher-forced distillation for improving negative signal quality. (3) *w/o $\mathcal{L}_{CRR}$* omits token-level collaborative prefer-

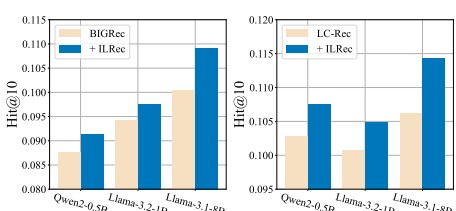

Figure 4: Performance Comparison w.r.t. Different Model Backbones on the Instrument dataset with BIGRec and LC-Rec.

ence adjustment (Equation 11), showing limited model performance and validating the importance of fine-grained collaborative signals. (4) *w/o $\mathcal{L}_{CPT}$* without both the $\mathcal{L}_{CPO}$ and $\mathcal{L}_{CPD}$. This variant results in a significant drop in model performances, verifying that both optimization and distillation loss are indispensable for achieving optimal performance. (5) *w/o Cross-layer Negative Signals (CNS)* directly extracts and penalizes negative signals in the final output logits. While keeping the distillation and reward modules, negative signals extracted from the ensemble of intermediate layers provide better optimization performance compared to those extracted from the final output layer. This verifies the rationality and effectiveness of leveraging intermediate layers for negative extraction. (6) *Final Layer (FL) & Intermediate Layers (IL) + $\mathcal{L}_{CPO}$* penalize negatives extracted from the final layer and intermediate layers respectively, without the distillation and reward modules. The results further verify that utilizing negatives from intermediate layers will achieve better performance and effectiveness rather than final layer's negatives.

## 3.4 FURTHER ANALYSIS

**Performance Comparison w.r.t. Numbers of Intermediate Layers.** ILRec relies on extracting fine-grained self-hard signals from multiple intermediate layers. To assess the impact of the layer count, we vary the number of intermediate layers from 0 (only final layer) to 4, with 0 layer indicating directly extracting and penalizing negative signals in the final output layer. As shown in Figure 5, using few layers yields limited gains due to insufficient usage of diverse and valuable negative signals encapsulated within different layers. Furthermore, incorporating too many lower layers degrades performance, likely due to their weak recommendation capabilities and introduction of noise. Thus, selecting an appropriate number of layers is crucial for optimal performance.

**Applying ILRec on Different Model Backbones.** To evaluate the generalizability of ILRec across different models, we apply ILRec to two relatively small but effective models, Llama-3.2-1B (Dubey et al., 2024) and Qwen2-0.5B (Yang et al., 2024), and train them on the Instrument dataset with BIGRec and LC-Rec paradigms respectively. The results shown in Figure 4 indicate that ILRec can consistently enhance recommendation performance across various models, highlighting the generalizability of our method.

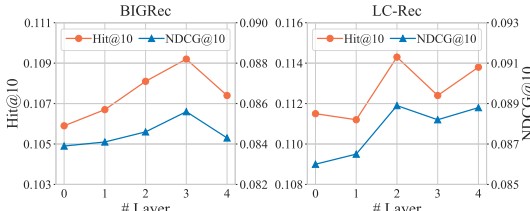 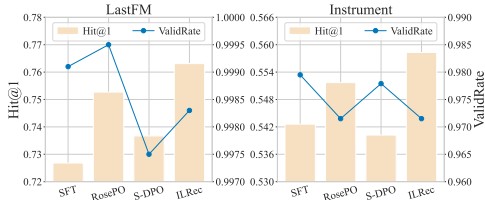

Figure 5: Performance Comparison w.r.t. Numbers of Intermediate Layers on the Instrument dataset with both BIGRec and LC-Rec paradigms.

Figure 6: Performance Comparison on Candidate-Ranking tasks on the LastFM and Instrument Datasets with different paradigms.

**Applying ILRec on Other Recommendation Tasks.** Apart from full ranking paradigms like BIGRec and LC-Rec, we evaluate ILRec on candidate ranking tasks, where LLMs select from a limited set of items, as in (Chen et al., 2024; Liao et al., 2024a). In this case, the negative space for LLM generation is relatively small, so that DPO-based methods (Chen et al., 2024; Liao et al., 2024a) tend to get a relatively stable performance rather than used in full ranking tasks. We follow the processed LastFM dataset provided in SDPO and also construct Instrument dataset in the same format. The results of our experiments are presented in Figure 6. We can see that ILRec still achieves higher performance in ranking-candidate tasks, indicating the effectiveness of our proposed method.

In addition, we conduct more detailed explorations, including applying different CF models (Appendix A.5), hyperparameter sensitivity analysis (Appendix A.7), efficiency analysis (Appendix A.6), and comparing different strategies for selecting intermediate layers (Appendix A.9).

## 4 RELATED WORK

**Sequential Recommendation.** Sequential recommendation aims to predict the next item for a user based on chronological interaction sequence. With the development of deep neural networks, early methods utilized complicated model architectures to better characterize user preferences, including convolutional neural networks (Tang & Wang, 2018), recurrent neural networks (Hidasi et al., 2016) and graph neural networks (Fan et al., 2019; Wu et al., 2022). More recently, self-attention and Transformer architectures (Vaswani, 2017) have been adopted for extracting implicit recommendation features, achieving improved performance (Kang & McAuley, 2018; Sun et al., 2019; Xie et al., 2022). Moreover, some studies also exploited pre-trained language models to enhance recommendation (Wu et al., 2021; Liu et al., 2023). In ILRec, we further optimize LLM-based recommendation systems on item semantic information and refine with collaborative signals from traditional models.

**LLMs for Recommendation.** Large language models (LLMs) have demonstrated remarkable capabilities in text understanding and generation. Existing studies combined LLMs with recommendation systems by leveraging LLMs to generate auxiliary information to enhance traditional recommendation models (Xi et al., 2024; Wei et al., 2024; Ren et al., 2024), or by simulating the virtual users in the recommendation environment (Wang et al., 2023; Zhang et al., 2024). Recently, using LLMs to recommend items directly has gained significant attentions. A widely adopted approach for recommendation adaptation is via fine-tuning paradigms (Bao et al., 2023b; Liao et al., 2024b), while some other studies attempted to optimize item representations and integrate collaborative information in LLM-based recommendation models (Liao et al., 2024b; Bao et al., 2024). Furthermore, to introduce negative samples in the training stage, recent work made usage of post-training methods, such as Direct Preference Optimization(DPO) (Rafailov et al., 2024), to align LLMs with user preferences (Liao et al., 2024a; Chen et al., 2024; Gao et al., 2024). However, DPO-based methods do not perform well especially as the negative spaces enlarge. We propose to leverage cross-layer fine-grained negative signals to enhance preference learning for LLM-based recommendation.

## 5 CONCLUSION

In this paper, we proposed **ILRec**, a novel fine-tuning framework to better align LLM-based recommender systems to user preference. Different from previous alignment tuning methods, we generated self-hard negatives from intermediate layers and incorporated them into SFT, which is both effective and efficient for adapting LLMs as recommender systems. We penalized the corresponding negative tokens by integrating fine-grained penalty coefficients into the cross-entropy loss. To enhance the

informativeness and reliability of provided negative signals, we also employed the output layer to supervise the token generation of intermediate layers. Additionally, we devised a collaborative reward regularization module to instill collaborative information and prevent potential false negatives from being overly penalized. Extensive experiments and in-depth analysis on three benchmarks demonstrated the superiority of our proposed ILRec framework. As future work, we aim to extend this fine-tuning approach to adapt LLMs to more diverse personalized tasks, while exploring more lightweight fine-tuning methods for efficient training.

## 6 REPRODUCIBILITY STATEMENT

All results presented in this work are fully reproducible. Implementation details and Hyperparameter selections are provided in Appendix A.4. The source code is available at https://anonymous.4open. science/r/ILRec-6FFE.

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

# A APPENDIX

## A.1 THE USE OF LARGE LANGUAGE MODELS

In accordance with the conference guidelines regarding the use of Large Language Models, we declare that LLMs were utilized solely as general-purpose assistive tools for language editing and polishing of the manuscript. No LLMs were involved in the design or development of research methods, experiments, or results. All scientific contributions, including methodological innovations, were conceived and executed entirely by the authors. The authors take full responsibility for the content of this paper.

## A.2 DATASETS STATISTICS

Table 3: Statistics of the processed datasets.

| Datasets | #Users | #Items | #Interactions | Sparsity |
|---|---|---|---|---|
| Instrument | 24,773 | 9,923 | 206,153 | 99.92% |
| Art | 45,142 | 20,957 | 390,832 | 99.96% |
| Game | 50,547 | 16,860 | 452,989 | 99.95% |

We conducted extensive experiments on three subsets of Amazon Review Data ((Ni et al., 2019)), *i.e.*, "*Musical Instruments*", "*Arts, Crafts and Sewing*" and "*Video Games*". Each of these datasets comprises user review data spanning from May 1996 to October 2018. For data preprocessing, we first remove unpopular users and items with less than five interactions through five-core filtering. Then, we create a historical interaction sequence sorted by timestamp for each. For fair comparison, the maximum item sequence length is uniformly set to 20 for all compared models. The statistics of datasets after preprocessing are shown in Table 3.

## A.3 BASELINE MODELS

We employ the following baselines:

**(1) Traditional sequential recommendation models**:

• **Caser** (Tang & Wang, 2018) is a method that modeling user behaviors through horizontal and vertical convolutional neural networks.

• **GRU4Rec** (Hidasi et al., 2016) is an RNN-based method that uses GRU to model the user behavior via encoding the item sequence.

- **SASRec** (Kang & McAuley, 2018) is the first sequential recommender based on the unidirectional self-attention mechanism.

**(2) LLM-based recommendation models**:

- **BIGRec** (Bao et al., 2023a) serves as an instruction-tuning LLM framework for sequential recommendations and generate recommended items based on embedding grounding.

- **LC-Rec** (Zheng et al., 2024) is a LLM-based sequential recommender that introduces semantic IDs to uniquely identify items.

- **SDPO** (Chen et al., 2024) introduces DPO into LRSs by sampling multiple negative items as rejected responses and incorporates a softmax loss over multiple negative samples.

- **RosePO** (Liao et al., 2024a) is a preference optimization framework that combines negative sampling strategies and personalized uncertainty to achieve fairness, unbiasedness, and robustness.

- **SPRec** (Gao et al., 2024) proposes a self-play fine-tuning method that consists of a SFT stage and a DPO stage in each training iteration, aiming at debiasing the preference alignment process.

## A.4 IMPLEMENTATION DETAILS

For all the LLM-based methods, we leverage Llama3.1-8B as the backbone LLM. For SFT-based baselines, we strictly follow the training settings of BIGRec (Bao et al., 2023a) and LC-Rec (Zheng et al., 2024) respectively. For DPO-based baselines, as these methods are not originally tested on all-ranking settings with the full datasets, we adjust the data format for the sake of rigorous comparison. Specifically, for RosePO (Liao et al., 2024a) and SDPO (Chen et al., 2024), we remove the list of candidate items in the prompt. We generate one self-hard negative samples from the SFT stage for RosePO, while randomly select 5 negative samples for SDPO. As for SPRec (Gao et al., 2024), as the dataset in our all-ranking settings are much larger than those sampled in SPRec, we set 3 iterations for training. For $\alpha$ in our method ILRec, which controls the threshold for selecting negative tokens, is tuned in the range $\{0.1, 0.5, 0.8, 1, 1.2\}$. For $\beta$ in ILRec, which controls the degree of penalization for each token, is tuned in the range $\{0.005, 0.01, 0.05, 0.1, 0.2\}$. For $\lambda$ and $\mu$, which serve as the coefficients for distillation loss and reward loss respectively, are tuned in the range $\{0.0005, 0.001, 0.005, 0.01, 0.05, 0.1\}$. For the collaborative model used in Section 2.4, we select SASRec (Kang & McAuley, 2018) to generate token-level reward. Given the high cost of tuning LLMs, we first identify the general scale of a hyper-parameter and then adjust it within a more limited range. For inference, we strictly follow the methods demonstrated in each paper. All experiments were carried out on eight A100 GPUs, each with 40GB of VRAM. We implement all traditional sequential recommendation models based on RecBole (Zhao et al., 2021). To ensure fair comparison, we set the embedding dimension of all models to 128 and obtain the best performance through hyperparameter grid search.

## A.5 PERFORMANCE COMPARISON W.R.T. DIFFERENT COLLABORATIVE FILTERING MODELS

Table 4: Performance Comparison w.r.t. Different Collaborative Filtering Models on the Instrument dataset.

| Methods | BIGRec | | LC-Rec | |
|---|---|---|---|---|
| | Hit@10 | NDCG@10 | Hit@10 | NDCG@10 |
| SASRec | **0.1091** | **0.0856** | **0.1143** | **0.0889** |
| GRU4Rec | 0.1085 | 0.0850 | 0.1136 | 0.0878 |
| BERT4Rec | 0.1089 | 0.0851 | 0.1142 | 0.0885 |
| Caser | 0.1073 | 0.0842 | 0.1125 | 0.0871 |
| BPR | 0.1068 | 0.0837 | 0.1125 | 0.0867 |

To verify the generalizability and effectiveness of utilizing collaborative filtering models in our approach, we further leverage some traditional collaborative filtering models to score tokens on the Instrument dataset. In details, we conduct experiments on SASRec (Kang & McAuley, 2018), GRU4Rec (Hidasi et al., 2016), BERT4Rec (Sun et al., 2019), Caser (Tang & Wang, 2018) and BPR. The results are shown in Tabel 4. These results indicate that, CF models with comparable standalone performance (e.g., SASRec, GRU4Rec, BERT4Rec) yield consistently similar improvements when

integrated into ILRec. This demonstrates that CRR does not depend on any specific CF models and remains stable improvement across strong CF models. We also observe a clear and consistent trend: higher-performing CF models (SASRec, GRU4Rec, BERT4Rec) bring larger gains in ILRec than relatively weaker CF models (Caser, BPR). This indicates that the effectiveness of chosen CF models is not heuristic or unpredictable, but rather correlates with the inherent quality of the CF model.

## A.6 EFFICIENCY ANALYSIS

Table 5: Efficiency of different Methods.

| Methods | Instrument | | | Art | | |
|---------|------|-------|--------|--------|-------|--------|
| | Time | Epoch | Sample | Time | Epoch | Sample |
| S-DPO | 7.25 h | 8(5 + 3) | 1 | 10.25 h | 8(5 + 3) | 1 |
| RosePO | 6.6 h | 8(5 + 3) | 1 | 9.53 h | 8(5 + 3) | 1 |
| SPRec | 7.8 h | 11(5 + 3 * 2) | 3 | 11.5 h | 11(5 + 3 * 2) | 3 |
| ILRec | 4.2 h | 5 | 0 | 7.46 h | 5 | 0 |

In this section, we further investigate the efficiency of the proposed method. As shown in Table 5, we demonstrate the training time, the number of training epochs (SFT + DPO * Iteration), and the number of sampling processes. Since ILRec integrate negative sampling and preference learning within the SFT Stage, there's no need for extra preference alignment processes or multiple forward calculation for different negative samples. The results demonstrate that ILRec does not introduce excessive training time costs compared to baseline methods, while achieving significant performance gains.

## A.7 HYPERPARAMETER SENSITIVITY ANALYSIS

Table 6: Hyperparameter Sensitivity Analysis with $\alpha$ and $\beta$ on Instrument and Art datasets.

| Hyperparameter | Instrument | | Art | |
|----------------|--------|---------|--------|---------|
| | Hit@10 | NDCG@10 | Hit@10 | NDCG@10 |
| baseline | 0.1062 | 0.0832 | 0.1045 | 0.0778 |
| $\alpha = 0.1$ | 0.1128 | 0.0882 | 0.1102 | 0.0799 |
| $\alpha = 0.5$ | 0.1138 | 0.0867 | 0.1084 | 0.0801 |
| $\alpha = 0.8$ | 0.1143 | 0.0889 | 0.1110 | 0.0807 |
| $\alpha = 1.0$ | 0.1115 | 0.0872 | 0.1118 | 0.0821 |
| $\alpha = 1.2$ | 0.1120 | 0.0883 | 0.1098 | 0.0799 |
| $\beta = 0.005$ | 0.1102 | 0.0857 | 0.1092 | 0.0795 |
| $\beta = 0.01$ | 0.1124 | 0.0880 | 0.1107 | 0.0799 |
| $\beta = 0.05$ | 0.1119 | 0.0881 | 0.1118 | 0.0821 |
| $\beta = 0.1$ | 0.1143 | 0.0889 | 0.1110 | 0.0813 |
| $\beta = 0.2$ | 0.1111 | 0.0869 | 0.1112 | 0.0807 |

We introduce some hyperparameters in ILRec. To evaluate the sensitivity of our method to these hyperparameters, we vary their values while keeping all other settings fixed and optimal, and observe the resulting impact on model performance. In Table 6, we present the effects of two key hyperparameters, $\alpha$ and $\beta$, which control the negative-signal selection penalization, on model performance across two datasets. These results indicate that ILRec achieves consistent improvements in a relatively stable range of hyperparameters, and achieves robustness of hyperparameter selection toward different datsets.

## A.8 DERIVING THE GRADIENT OF CROSS-LAYER PREFERENCE OPTIMIZATION LOSS

We derive the gradient of cross-layer preference optimization loss as follows:

$$\nabla_\theta \mathcal{L}_{CPO} = -\nabla_\theta \log \frac{\exp(p(y_t))}{\sum_{v \in \mathcal{V}} \exp(p(v)(1 + \beta w_v))}$$

$$= -\nabla_\theta p(y_t) + \nabla_\theta \log \sum_{v \in \mathcal{V}} \exp(p(v)(1 + \beta w_v))$$

$$= -\nabla_\theta p(y_t) + \frac{\nabla_\theta \sum_{v \in \mathcal{V}} \exp(p(v)(1 + \beta w_v))}{\sum_{v \in \mathcal{V}} \exp(p(v)(1 + \beta w_v))}$$

$$= -\frac{\sum_{v' \in \mathcal{V}_N} \exp(p(v')(1 + \beta w_{v'}))}{\sum_{v \in \mathcal{V}} \exp(p(v)(1 + \beta w_v))} \nabla_\theta p(y_t) + \sum_{v' \in \mathcal{V}_N} [\frac{(1 + \beta w_{v'}) \exp(p(v')(1 + \beta w_{v'}))}{\sum_{v \in \mathcal{V}} \exp(p(v)(1 + \beta w_v))}] \nabla_\theta p(v'),$$

in which $\mathcal{V}_N$ refers to all the tokens without the ground-truth token $y_t$. To penalize high probability tokens provided by intermediate layers, ILRec assigns the gradient of each negative token $v'$ an extra weighting $\frac{(1 + \beta w_{v'}) \exp(p(v')(1 + \beta w_{v'}))}{\sum_{v \in \mathcal{V}} \exp(p(v)(1 + \beta w_v))}$. For a challenging high probability token $v'$ provided by intermediate layer, the corresponding probability $w_{v'}$ for token $v'$ is also large, leading to larger extra weighting and more decline in the likelihood of generating specific $v'$.

## A.9 STRATEGIES FOR SELECTING INTERMEDIATE LAYERS

### A.9.1 PERFORMANCE COMPARISON W.R.T. DIFFERENT LAYER SELECTION STRATEGIES

To ensure informative negatives, ILRec selects $k$ consecutive intermediate layers starting from the final layer as the negative generator. To further justify this design choice, we conduct additional experiments comparing our strategy (consecutive layers 31–29) with several alternative layer-selection approaches: (1) three randomly chosen non-consecutive layers from 31–24, (2) individual layers 31, 30, and 29, (3) medium layers 26–24, and (4) shallow layers 21–19. The results are presented in Table 7.

Table 7: Performance Comparison w.r.t. Different Layer Selection Strategies on the Instrument Dataset using BIGRec training paradigm.

| Methods | Hit@10 | NDCG@10 |
|---|---|---|
| ILRec | **0.1091** | **0.0856** |
| Non-consecutive Layers | 0.1084 | 0.0843 |
| Medium Layers | 0.1043 | 0.0820 |
| Shallow Layers | 0.1037 | 0.0813 |
| $L_{31}$ | 0.1067 | 0.0839 |
| $L_{30}$ | 0.1080 | 0.0844 |
| $L_{29}$ | 0.1075 | 0.0836 |

As shown in Table 7, our consecutive deep-layer selection consistently outperforms all alternative strategies. Deeper intermediate layers capture stronger task-specific semantics, producing more reliable and informative negatives. In contrast, shallower layers exhibit weaker predictive capability, leading to noisier negatives that hinder ILRec's effectiveness. Strategies based on non-consecutive or shallow layers tend to mix in less informative layers without a principled selection rule, while relying on a single layer reduces stability and limits the diversity of negative signals—both contributing to degraded performance.

### A.9.2 THE GUIDANCE OF SELECTING $k$ ACROSS DIFFERENT SCALES OF BACKBONES.

When applying ILRec to models of different sizes, we conduct experiments to examine how the choice of $k$ affects performance across various model scales. Based on these results, we provide practical guidance on selecting the optimal range of $k$ for different model sizes, helping avoid repeated and costly hyperparameter tuning. Specifically, while keeping all other hyperparameters fixed, we varied $k$ from 1 to 7 and evaluated different models' performance (Llama3.2-1B, Llama3.2-3B and Llama3.1-8B) with the ILRec task on the Instrument dataset, using Hit@10 as the main metric. The results are summarized as follows:

Table 8: Performance Comparison w.r.t. Different $k$ among Different Model Backbones on Amazon Instrument.

| Methods | 1 | 2 | 3 | 4 | 5 | 6 | 7 |
|---------|-----|-----|-----|-----|-----|-----|-----|
| Llama3.2-1B | 0.0972 | **0.0976** | 0.0968 | 0.0950 | 0.0953 | 0.0938 | 0.0947 |
| Llama3.2-3B | 0.0993 | 0.1019 | **0.1024** | 0.1022 | 0.1008 | 0.1013 | 0.1002 |
| Llama3.1-8B | 0.1067 | 0.1081 | **0.1091** | 0.1084 | 0.1088 | 0.1075 | 0.1057 |

From Table 8, we can summarize the guidance for selecting $k$ for different model backbones as follows:

• **The optimal $k$ consistently lies within intermediate layers closed to the final layer($\leq 5^{th}$ layer)**, as shown in the bold data in Table 8. These deep layers have sufficient predictive capability to provide informative negatives, compared with shallower layers.

• **ILRec is robust to a range of nearby $k$ values**, as shown in the underlined data in Table 8. Performance around the optimal $k$ varies smoothly, and neighboring values also yield comparable gains, meaning that only a small range of $k$ needs to be tested to achieve stable performance improvements.

• **Larger models support a broader effective range of $k$.** Intermediate layers in larger models generate stronger predictions and higher-quality negative signals. Therefore, while 1B models have a narrow effective range of $k$ for achieving stable performance improvement, 3B and 8B models benefit from a broader selection of $k$.

