# OpenReview forum: "Intermediate Layers Can Be Self-Hard Negative Generator For Large Language Model Based Recommendation"
_ICLR.cc/2026/Conference — Submitted to ICLR 2026_

### Official Review · Reviewer_UTWL · 2025-10-28

**Soundness:** 2
**Presentation:** 3
**Contribution:** 2
**Rating:** 4
**Confidence:** 4

**Summary:**

This paper proposes ILRec, a fine-tuning framework for LLM-based recommender systems that leverages self-hard negative signals extracted from intermediate layers. The key idea is that intermediate layers can act as “non-expert” models whose outputs provide fine-grained, dynamically generated negatives during training. ILRec integrates these negatives through three mechanisms: (1) Cross-Layer Preference Optimization (CPO), which penalizes high-probability negatives in the final layer’s logits; (2) Cross-Layer Preference Distillation (CPD), which transfers knowledge from the final layer to intermediate layers to enhance negative informativeness; and (3) Collaborative Reward Regularization (CRR), which uses a lightweight CF model to prevent over-penalization of false negatives. Experiments on Amazon Review datasets demonstrate consistent improvements over existing LLM-based baselines such as BIGRec, LC-Rec, and DPO-style methods

**Strengths:**

1. The work targets a critical challenge in LLM-based recommendation: how to efficiently and adaptively introduce effective negative samples during fine-tuning.

2. The paper is well-structured and easy to follow.

3. The idea of extracting negative signals directly from intermediate layers is interesting.

**Weaknesses:**

1. While the idea of extracting negatives from intermediate layers is interesting, its distinction from simply selecting high-probability tokens from the final layer’s distribution (or using beam search self-generated negatives) is not fully clear. Both intuitively capture similar model uncertainty patterns. The paper would benefit from deeper analysis and ablation (e.g., comparing intermediate vs. final-layer negatives, or evaluating informativeness across layers).

2. The comparison set omits several recent and relevant RLVR-based LLM recommendation methods, which represent the current frontier. The paper primarily compares against DPO-style preference optimization (e.g., RosePO, SDPO, SPRec), which, as the authors acknowledge, suffer from policy-gap issues. Including RLVR-based baselines would strengthen claims of superiority.

3. The inclusion of modules like Collaborative Reward Regularization (CRR) adds complexity, yet the reported gain appears modest. A clearer justification or simplification could make the approach more elegant.

4. The authors argue that sequence-level reward is a limitation of existing methods, but provide neither theoretical reasoning nor empirical evidence to substantiate this claim.

**Questions:**

1. Equation (4) introduces α as a key hyperparameter controlling the threshold for selecting negative tokens. How is this parameter set in practice, and how sensitive are results to its value? A more systematic sensitivity analysis would improve reproducibility and interpretability.

2. The paper averages logits from multiple layers (Equation 3) to form ensemble logits. Have the authors compared this averaging scheme with alternative intra-model metrics (e.g., maximum probability, variance, entropy-based weighting)? Averaging may blur layer-specific diversity.

3. The motivation cites the idea that “expert models can be optimized by contrasting them with non-expert models.” Have the authors considered using an external collaborative recommender (e.g., SASRec) as the non-expert model to provide complementary negatives? This could be an interesting extension or ablation to demonstrate the generality of the proposed contrastive mechanism.

---

> ### Author Response · Authors · 2025-11-27
> **Rebuttal by Authors**
>
> Thank you for your careful evaluation and helpful suggestions! We appreciate the opportunity to clarify several aspects of our method, and we address each of your comments in the following response.
>
> ---
>
> ## Response to W1 part 1/2
>
> Thank you for your positive assessment and for finding our method interesting! To more clearly demonstrate the effectiveness of ILRec’s use of intermediate-layer negatives, we provide an expanded comparison and a more in-depth analysis contrasting negatives from intermediate layers and those from the final layer.
>
> 1. **Comparison Experiments**
>
> To compare ILRec with the method that directly utilizes negatives from the final layers, we have already provided the comparison results in our paper in Table 2, in which we add $\mathcal{L}_ {CRR}$ and $\mathcal{L}_ {CPD}$ when utilizing final layer's negatives to ensure a fair comparison with ILRec. To enable a more comprehensive comparison, we conduct expriments that omit distillation and collaborative loss, only keep $\mathcal{L}_ {CPO}$ to penalize negative tokens for these two methods. All the results are listed in the following table. In the table, **FL** denotes sampling negatives from the **F**inal **L**ayer, and **IL** denotes sampling negatives from the **I**ntermediate **L**ayers. We also indicate the training loss applied for each configuration alongside FL and IL.
>
> **Table 1: Performance comparison between Final layer negatives and Intermediate layers negatives on Amazon Instrument.**
>
> | Method                              | BIGRec |        | LCRec  |        |
> | ----------------------------------- | ------ | ------ | ------ | ------ |
> |                                     | H@10    | N@10    | H@10    | N@10    |
> | SFT                                 | 0.1004 | 0.0799 | 0.1062 | 0.0832 |
> | FL + $\mathcal{L}_ {CPO}$                     | 0.1029 | 0.0814 | 0.1097 | 0.0844 |
> | IL + $\mathcal{L}_ {CPO}$                | 0.1048 | 0.0827 | 0.1118 | 0.0863 |
> | FL + $\mathcal{L}_ {CPO}$ + $\mathcal{L}_ {CPD}$ + $\mathcal{L}_ {CRR}$ | 0.1059 | 0.0839 | 0.1115 | 0.0866 |
> | IL + $\mathcal{L}_ {CPO}$ + $\mathcal{L}_ {CPD}$ + $\mathcal{L}_ {CRR}$                              | 0.1091 | 0.0856 | 0.1143 | 0.0889 |
>
> The results above demonstrate that **utilizing negatives from intermediate layers will achieve better performance and effectiveness rather than final layer's negatives.**
> Below we will give a detailed analysis and comparisons between negatives extracted from the final layer and from intermediate layers.

---

> ### Author Response · Authors · 2025-11-27
> **Rebuttal by Authors**
>
> ## Response to W1 part 2/2
>
> 2. **Detailed Analysis**
>
> To further clarify and analyze why using negatives from the final layer underperforms compared to intermediate-layer negatives in our experiments, we conduct additional studies and summarize our findings as follows:
>
> - **Negatives from the final layer are more likely to be false negatives or overly hard negatives**, which can interfere with the model’s learning process.
>
> - **The SFT loss already penalizes hard negatives at the final layer**, so applying additional penalties to these negatives provides only marginal improvements.
>
> (1) The final layer exhibits substantially stronger recommendation capability than intermediate layers. Consequently, high-probability negatives extracted from the final layer often correspond to items that the model implicitly considers potentially relevant to the user. In other words, these samples are more prone to being false negatives. Aggressively penalizing such items can distort the learned user preference representation and lead to degraded performance.
>
> To further validate that **negatives from the final layer are more likely to be false negatives compared with those from intermediate layers**, we employed two external models—a semantic matching model (E4SRec[1]) and a collaborative filtering model (SASRec)—to measure the similarity between sampled items and their corresponding users from semantic and collaborative perspectives.
>
> Specifically, we randomly selected 3 sets of test samples in the Instrument Dataset, each with 50 test data samples. Then, we beam search intermediate layers and the final layer to generate their top-5 non-positive predicted items over each test data, respectively. After sampling, we computed user–item similarity scores of these items using both semantic and CF models and averaged the results over three test sample sets.
>
> **Table 2: User-Item similarity comparison between Final layer negatives and Intermediate layers negatives.**
>
> |                          | Final Layer | Intermediate Layers |
> | ------------------------ | ----------- | ------------------- |
> | Semantic Model（E4SRec） | **0.8232**      | 0.7814              |
> | CF Model（SASRec）       | **0.7559**      | 0.7380              |
>
> Across both semantic and collaborative similarity metrics, **the non-positive predictions from the final layer are more likely to be recommended to the user than those from intermediate layers**. This indicates that **final-layer negatives tend to lie closer to the user’s preference distribution and are therefore more likely to be false negatives**.
>
> （2）Since ILRec directly applies additional penalties to negative tokens during the SFT stage, we analyzed the gradient of LLM’s SFT loss as follows (we set temperature as 1 for simplicity):
>
> $$
>     \nabla_ \theta \mathcal{L}_ {SFT} = - \nabla_ \theta \log \frac{\exp(p(y_t))}{\sum_ {v\in \mathcal{V}} \exp(p(v))}
> $$
>
> $$
>     = - \frac{\sum_ {v' \in \mathcal{V}_ N}\exp (p(v'))}{\sum_ {v \in \mathcal{V}} \exp (p(v))} \nabla_ \theta p(y_t) + \sum_ {v' \in \mathcal{V}_ N}[\frac{\exp (p(v'))}{\sum_ {v \in \mathcal{V}} \exp (p(v))}] \nabla_ \theta p(v'),
> $$
>
> in which $\mathcal{V}_ N$ refers to all the tokens without the ground-truth token $y_t$. In $\mathcal{L}_ {sft}$, **the model already imposes penalties on hard negative tokens at the final layer**. The gradient of each negative token $v'$ is assigned with an extra weighting $\frac{\exp (p(v'))}{\sum_{v \in \mathcal{V}}\exp (p(v))}$, with stronger penalties applied to harder negatives with higher $p(v')$.
>
> Also, as shown in Table 1, **applying extra penalties to these hard negatives in the final-layer loss (**FL** + $\mathcal{L}_{CPO}$ line) yields only marginal improvements compared with the original SFT baseline (**SFT** line)**. In contrast, ILRec outperforms the variant that uses final-layer negatives in both settings—with and without the distillation loss applied to further enhance negative signals—demonstrating the effectiveness and robustness of ILRec’s design.
>
> [1]Li, Xinhang, et al. "E4srec: An elegant effective efficient extensible solution of large language models for sequential recommendation." arXiv preprint arXiv:2312.02443 (2023).

---

> ### Author Response · Authors · 2025-11-27
> **Rebuttal by Authors**
>
> ## Response to W2
>
> Thank you for raising this insightful point. We appreciate the reviewer’s suggestion regarding RLVR-based baselines. To provide further clarification, we will demonstrate the difference between ILRec and other RLVR-based methods and inplement experiments for detailed comparison.
>
> Our work primarily focuses on **improving negative sample mining** and **leveraging these signals during the SFT stage**, targeting direct item generation tasks in recommendation. In contrast, recent RLVR frameworks for recommendation[1,2] emphasize **explicit reward design** and **reward-triggered reasoning processes**, which are typically applied to reason-then-generate recommendation tasks, including both explicit reasoning and latent reasoning.
>
> **We appreciate the reviewer’s suggestion that the RLVR paradigm can help address policy gap issues.** Therefore, to explore this, we conduct experiments to compare the effectiveness of ILRec and $R^2ec$ [1] on the Instrument dataset using BIGRec training paradigm. The results are as follows:
>
> **Table 3: Performance comparison between $R^2ec$ and ILRec.**
>
> | Method | H@5    | N@5    | H@10   | N@10   | Training Time |
> | ------ | ------ | ------ | ------ | ------ | ------------- |
> | SFT    | 0.0786 | 0.0742 | 0.1004 | 0.0799 | 3.8h          |
> | ILRec  | **0.0844** | **0.0788** | **0.1091** | **0.0856** | 4.1h          |
> | $R^2ec$   | 0.0827 | 0.0763 | 0.1062 | 0.0835 | 10.6h         |
>
>
> Attributed to the **fine-grained negative sampling** and **the continuous refinement of negative sample quality** from intermediate layers, ILRec achieves consistently strong improvements. Moreover, since ILRec operates entirely within the SFT stage—without requiring additional preference alignment or rollout computations—it achieves significantly shorter training time, as shown in Table 3.
>
> Therefore, ILRec serves as a **lightweight yet effective** training paradigm for LLM-based recommendation systems, especially under limited computational resources or constrained training time.
>
> [1] You R, Li Y, Lin X, et al. R
>  ec: Towards Large Recommender Models with Reasoning[C]//The Thirty-ninth Annual Conference on Neural Information Processing Systems. 2025.
>
> [2] Zhang Y, Xu W, Zhao X, et al. Reinforced Latent Reasoning for LLM-based Recommendation[J]. arXiv preprint arXiv:2505.19092, 2025.

---

> ### Author Response · Authors · 2025-11-27
> **Rebuttal by Authors**
>
> ## Response to W3
>
> Thank you for raising this important point! Below, we offer a more detailed explanation of why the CRR module is essential to ILRec.
>
> While the performance improvement brought by CRR may seem relatively small, CRR plays a pivotal role in **stabilizing ILRec by providing collaborative-filtering guidance and preventing the model from excessively penalizing items that are potential false negatives**.
>
> To more systematically verify its necessity, we examine the impact of CRR from three different perspectives:
>
> **(1) Statistical significance test**
>
> To verify that the improvement from the CRR module is robust, we ran each experiment with 15 different random seeds and performed a paired t-test comparing ILRec with and without CRR. Across all metrics, the improvements are **statistically significant (p < 0.01)**, indicating that the gains from CRR are consistent and are not outcomes with random training fluctuations.
>
> **(2) Additional dataset evaluation demonstrates higher gains**
>
> To examine whether CRR’s effectiveness generalizes beyond a single type of dataset, we further evaluate ILRec on additional datasets: Goodreads and Yelp.
>
> **Table 4: Performance of CRR module on Yelp and Goodreads.**
>
> |               | Yelp         |              | Goodreads    |              |
> | ------------- | ------------ | ------------ | ------------ | ------------ |
> |               | H@10          | N@10          | H@10          | N@10          |
> | ILRec         | 0.0445 +4.2% | 0.0323 +3.5% | 0.0681 +1.9% | 0.0537 +1.5% |
> | ILRec w/o CRR | 0.0427       | 0.0312       | 0.0668       | 0.0529       |
>
>
> Results show that the CRR module leads to **larger improvements on Yelp (+4.2% and +3.5% for Hit@10 and NDCG@10)**, and delivers consistent gains across Goodreads. This shows that CRR is effective across multiple datasets, consiH zstently improving performance, and even yielding notably larger gains on certain datasets.
>
> **(3) Case study: how CRR mitigates the penalization of potential false negatives**
>
> We further conduct a case study to investigate how CRR improves ILRec’s performance. Our approach is to **select test samples that require capturing collaborative filtering (CF) signals and compare the performance of ILRec with and without the CRR module on these samples.** This analysis allows us to **evaluate whether CRR effectively incorporates CF information and helps prevent the model from over-penalizing potential false negatives from the perspective of CF models**.
>
> Specifically, we train the following 4 models on the Art dataset using LC-Rec training paradigm at first:
>
> - a CF model (SASRec) $M_{cf}$
> - a SFT model (LC-Rec) $M_{sft}$
> - ILRec without CRR $M_{wocr}$
> - ILRec with CRR $M_{cr}$
>
> Then, we use $M_{cf}$ to select 40 test data samples in the Art dataset, in which $M_{cf}$ achieves Hit@10=1 among these samples
> Finally, we evaluate the remaining three models on these samples. For each model, we report Hit@10 at three checkpoints, selected sequentially according to the model’s training process. The results are summarized below:
>
> **Table 5: Performancce comparison of different models on CF-guided test sets.**
>
> |  Model          | Origin | Ckpt-1 | Ckpt-2 | Ckpt-3 |
> | ---------- | ------ | ------ | ------ | ------ |
> | $M_{sft}$  | 0.05   | 0.3    | 0.4    | 0.425  |
> | $M_{wocr}$   | 0.05   | 0.25   | 0.4    | 0.325  |
> | $M_{cr}$ | 0.05   | 0.425  | 0.525  | 0.525  |
>
>
> These results show that ILRec with CRR consistently captures items favored by CF models, while ILRec without CRR tends to over-penalize these potential false-negative items and even performs worse than the original $M_{sft}$.

---

> ### Author Response · Authors · 2025-11-27
> **Rebuttal by Authors**
>
> ## Response to W4
>
>
> Thank you for raising this important question. Below, we provide a consolidated justification from theretical reasoning to empirical evidence, showing that token-level preference optimization would be better than sequence-level method in LLM-based recommendation.
>
> **(1) Theoretical reasoning: why sequence-level optimization is problematic**
>
> Recent studies on DPO and preference learning (e.g., TIS-DPO[1], Step-DPO[2]) have explicitly analyzed the drawbacks of sequence-level reward modeling, which can be summarized as two issues:
>
> - **Credit assignment issue**: DPO treats all tokens in the positive or negative sequences uniformly and fails to generate fine-grained gradients or rewards to guide the optimization of tokens or steps within the sequence. In some cases, overlapping tokens between positive vs. negative sequences yield incorrect gradient signals[1].
>
> - **Ineffective usage of negatives**: When the negative space is large, sequence-level methods struggle to efficiently learn fine-grained token generation patterns from a limited number of positive–negative sequence comparisons.
>
> Recent work has therefore shifted toward token-level preference modeling and step-wise comparisons [1,2] to address these limitations.
>
> For LLM-based recommendation, the challenge is even greater: the negative sample space is extremely large, and every generated token contributes to the final item prediction, demanding fine-grained supervision at the token level.
>
> To tackle this, ILRec introduces **token-level positive–negative comparison during generation and modulates the penalty on negatives using both intermediate-layer probabilities and signals from a CF model**. This mechanism provides more precise credit assignment and greatly improves scalability under large negative sample spaces.
>
>
> **(2) Empirical evidence: token-level optimization is more effective**
>
> To verify these theoretical insights, we explicitly compare **token-level DPO** vs. **sequence-level DPO** using LC-Rec training paradigm:
> - For sequence-level DPO, we use beam search to generate top-3 sequence-level negatives from the model after SFT.
> - For token-level DPO, we sample 3 negative tokens with high probabilities at each step following the ground-truth token from the model after SFT.
>
> These samples were then used for subsequent DPO training, using traditional DPO loss and step-DPO loss in [2] (in which each token are considered as a step in LLM-based recommendation).
>
> **Table 6: Performance comparison between token-level DPO and sequence-level DPO on Amazon Instrument.**
>
> | Method         | H@5    | N@5    | H@10   | N@10   |
> | -------------- | ------ | ------ | ------ | ------ |
> | Token-level    | **0.0912** | **0.0784** | **0.1071** | **0.0820** |
> | Sequence-level | 0.0858 | 0.0752 | 0.1038 | 0.0795 |
>
>
> The results clearly show the superiority of the token-level approach. ILRec builds upon this insight by directly extracting high-quality and fine-grained negatives from intermediate layers during SFT and optimizing the negative-sample utilization process, leading to significantly improved model performance.
>
> [1] Liu, Aiwei, et al. "Tis-dpo: Token-level importance sampling for direct preference optimization with estimated weights." arXiv preprint arXiv:2410.04350 (2024).
> [2] Lai, Xin, et al. "Step-dpo: Step-wise preference optimization for long-chain reasoning of llms." arXiv preprint arXiv:2406.18629 (2024).

---

> ### Author Response · Authors · 2025-11-27
> **Rebuttal by Authors**
>
> ## Response to Q1
>
>
> Thank you for raising this point regarding the choice and sensitivity of the hyperparameter $\alpha$. We clarify its role and provide systematic analysis below.
>
> **(1) Practical role of $\alpha$**
>
> We use $\alpha$ to dynamically select negative tokens whose predicted probability at each intermediate layer exceed $\alpha p_{y_{t}}$. Therefore, ILRec can identify hard negatives whose likelihood is comparable to or more than that of the positive token, while dynamically selecting negatives according to the prediction accuracy of the current logits.
>
> **(2) Sensitivity analysis**
>
> As reported in Table 6 (Appendix A.7) in our paper, we performed a systematic hyperparameter study over both $\alpha$ and $\beta$ on two benchmark datasets.
> For clarity, we demonstrate the results of $\alpha$ within our tuning range as follows:
>
> **Table 7: Hyperparameter sensitivity analysis for $\alpha$.**
>
> | Hyperparameter | Instrument Hit@10 | Instrument NDCG@10 | Art Hit@10 | Art NDCG@10 |
> | -------------- | ----------------- | ------------------ | ---------- | ----------- |
> | **baseline**   | 0.1062            | 0.0832             | 0.1045     | 0.0778      |
> | **α = 0.1**    | 0.1128            | 0.0882             | 0.1102     | 0.0799      |
> | **α = 0.5**    | 0.1138            | 0.0867             | 0.1084     | 0.0801      |
> | **α = 0.8**    | **0.1143**            | **0.0889**             | 0.1110     | 0.0807      |
> | **α = 1.0**    | 0.1115            | 0.0872             | **0.1118**     | **0.0821**      |
> | **α = 1.2**    | 0.1120            | 0.0883             | 0.1098     | 0.0799      |
>
>
> Across both datasets, ILRec exhibits stable performance when **$\alpha$ ∈ {0.8, 1.0, 1.2}**.
> This indicates that the model is **not highly sensitive to α within specific range.**
> The best-performing settings are $\alpha$ = 0.8 for the Instrument Dataset and $\alpha$ = 1.0 for the Art Dataset.
>
> **(3) Recommended practical settings**
>
> In practical applications for other datasets or recommendation tasks, we prefer selecting $\alpha$ within the range {0.8, 1.0}, since values in this interval reliably help the model distinguish negative tokens whose predicted probability is higher than—or very close to—that of the positive token at intermediate layers. This makes the dynamic negative sampling mechanism effective and robust across different settings.
>
> ---
>
> ## Response to Q2
>
> Thank you for the insightful suggestion for aggregating intermediate-layer logits using alternative intra-model metrics!
>
> We have conducted additional experiments comparing our averaging scheme with several alternative intra-model metrics you've mentioned, including maximum-probability, variance-based weighting, and entropy-based weighting. We utilize these metrics as layer-specific weight when aggregating intermediate layers.The results are demonstrated as follows:
>
> **Table 8: Performance comparison with different intro-model metrics on Amazon Instrument.**
>
> |              | BIGRec |        | LCRec  |        |
> | ------------ | ------ | ------ | ------ | ------ |
> |              | H@10    | N@10    | H@10    | N@10    |
> | mean average | **0.1091** | **0.0856** |0.1143 | 0.0889 |
> | Variance     | 0.1074 | 0.0840 | 0.1130 | 0.0862 |
> | Entropy      | 0.1066 | 0.0833 | **0.1149** | **0.0893** |
> | Maximum-probability         | 0.1061 | 0.0836 | 0.1112 | 0.0841 |
> | SFT | 0.1004 | 0.0799 | 0.1062 | 0.0832 |
>
>
>
> Our results show that **all these aggregation methods yield relatively stable performance improvement**, with entropy-based weighting even slightly outperforming our logits-averaging method in some cases. **This confirms that more sophisticated and appropriate layer-specific or intra-model aggregation strategies are indeed a valuable direction for future work**！
>
> Nevertheless, in our current settings, **directly averaging intermediate-layer logits has already offered a simple, efficient, and consistently effective solution.**
> - Firstly, under the ILRec framework, which **co-optimizes negative-sample generation and utilization**, each intermediate layer can still gradually learn and preserve diverse layer-specific information through distillation.
> - Secondly, alternative aggregation strategies based on **metrics such as variance or entropy currently lack a clear and consistent correlation with the quality of generated negative samples**. In contrast, the simple averaging scheme provides a robust, efficient, and effective way to integrate logits.

---

> ### Author Response · Authors · 2025-11-27
> **Rebuttal by Authors**
>
> ## Response to Q3
>
> Thank you for your thoughtful perspectives, which we think could highlight a broader search direction on weak-to-strong learning! We will first **clarify the criteria for selecting an effective non-expert model**, and then **analyze ILRec’s performance when using SASRec-generated negatives for comparison**.
>
> **(1) Not all weak models are informative weak models.**
> While an expert model can indeed learn by contrasting itself with a weaker model, only those non-expert models whose decision boundaries remain partially aligned with the expert can provide meaningful negative signals. If the non-expert is too weak or too misaligned, its predicted negatives become nearly random and are not informative enough, which limits their usefulness and may even introduce noise that disrupts expert model training. This is also the reason why we **select intermediate layers closed to the final layer for negative generation**, and **design distillation module for each layers to keep the quality of negatives**.
>
> **(2) Empirical examination: SASRec as a non-expert model.**
> To directly study whether an external collaborative filtering model (SASRec) can serve as an effective non-expert, we removed ILRec’s CRR and CPD modules as the baseline. We computed SASRec’s predicted scores following Eq. (10) in our paper and selected SASRec’s top-scoring tokens as negative candidates to penalize in the final layer. Results on BIGRec and LC-Rec are shown below:
>
> **Table 9: Performance comparison with different sources of negatives.**
>
> | Source of Negatives | BIGRec |        | LCRec  |        |
> | ------------------- | ------ | ------ | ------ | ------ |
> |                     | H@10    | N@10    | H@10    | N@10    |
> | None（SFT）         | 0.1004 | 0.0799 | 0.1062 | 0.0832 |
> | SASRec              | 0.1018 | 0.0804 | 0.1078 | 0.0836 |
> | Intermediate layers | **0.1048** | **0.0827** | **0.1118** | **0.0863** |
>
> Based on the experimental results, we summarize the following two observations:
> -  **Using SASRec negatives yields improvements over the SFT baseline**, indicating that SASRec does provide a limited amount of valuable CF-driven error patterns for the LLM to learn from.
> -  **SASRec negatives still fall short of ILRec’s intermediate-layer negatives**, suggesting that SASRec’s ranking signals are less aligned with the LLM’s token-generation process and are therefore less capable of providing high-quality token-level negatives for the expert layer.
>
> We appreciate the reviewer’s insightful perspective, and we believe this highlights a broader research direction:
> **how to systematically select or ensemble multiple weak models under the weak-to-strong paradigm to construct more informative and complementary negative signals for strong LLM recommenders**. We agree this is a highly promising direction and plan to further explore it in future work.

---

### Official Review · Reviewer_t9vT · 2025-10-30

**Soundness:** 2
**Presentation:** 3
**Contribution:** 2
**Rating:** 4
**Confidence:** 4

**Summary:**

The paper identifies that existing preference optimization methods for LLM-based recommender systems suffer from two major limitations: (1) they assign only coarse-grained rewards to entire sequences, and (2) they sample negative items in an offline manner, both of which lead to suboptimal performance.

To solve these problems, the paper proposes **ILRec**, a fine-tuning framework for LLM-based recommender systems that extracts self-hard negative tokens for fine-grained optimization. Specifically, ILRec ensembles the logits from the intermediate layers and selects the negative tokens with high logit values as the self-hard negative tokens, which are then emphasized during optimization. ILRec also incorporates a distillation loss to improve the recommendation ability of intermediate layers and a collaborative reward regularization to inject collaborative information. Extensive empirical results demonstrate that ILRec outperforms prevalent preference optimization methods and is effective across different backbone models and item representations.

**Strengths:**

**S1: Well-founded Motivation.** The motivation is well-founded. Since the quality of negative items is important to recommendation, it is crucial to improve the suboptimal offline negative sampling pattern adopted by current preference optimization methods.

**S2: Excellent Generality.** ILRec is effective for both textual item representation (BigRec [1]) and semantic IDs (LC-Rec). In addition, ILRec also enhances the performance across different backbone models and collaborative models.

[1] Bao K, Zhang J, Wang W, et al. A bi-step grounding paradigm for large language models in recommendation systems[J]. ACM Transactions on Recommender Systems, 2025, 3(4): 1-27.
[2 ]Zheng B, Hou Y, Lu H, et al. Adapting large language models by integrating collaborative semantics for recommendation[C]//2024 IEEE 40th International Conference on Data Engineering (ICDE). IEEE, 2024: 1435-1448.

**Weaknesses:**

**W1: Method Design.** In some previous works [1], the introduction of weak experts is motivated by the lack of supervision from stronger experts. However, since the final layer generally performs better than the intermediate layers, the negative tokens derived from the logits of the final layer can provide stronger supervision, suggesting that incorporating intermediate layers may not necessarily lead to better results.  Although the ablation studies show that using negative signals from the last layer results in suboptimal results, the distillation loss should be removed in this setting because no intermediate layers are introduced, which means there is no need to enhance them. Additional discussions on these two methods, along with experiments introducing negative tokens from the final layer **without applying** the distillation loss, could more clearly validate the effectiveness of ILRec.

**W2: Loss Analysis.** It is not straightforward to derive from the loss function of CPO in Equation 7 that challenging negative signals will be penalized more. Thus, a gradient analysis on CPO loss, similar to that presented in [2], is recommended to better illustrate the feature of CPO loss.

**W3: Ablation Studies.** As shown in Table 2, incorporating $\mathcal{L}_{\text{CRR}}$ yields only marginal relative improvements (less than 2%), which cannot fully demonstrate the effectiveness of the collaborative reward regularization. Additional explanations and experiments are needed to further justify the importance of the collaborative signals.



[1] Sang J, Wang Y, Zhang J, et al. Improving weak-to-strong generalization with scalable oversight and ensemble learning[J]. arXiv preprint arXiv:2402.00667, 2024.

[2] Chen Y, Tan J, Zhang A, et al. On softmax direct preference optimization for recommendation[J]. Advances in Neural Information Processing Systems, 2024, 37: 27463-27489.

**Questions:**

**Q1: Implementation Details.** Since LLM-based recommenders are prone to generating invalid or out-of-vocabulary items, it is unclear how the model in this paper performs inference for the full-ranking task. Please clarify the inference strategy and specify the exact evaluation pipeline adopted during testing.

---

> ### Author Response · Authors · 2025-11-27
> **Rebuttal by Authors**
>
> Thank you for your constructive feedback and valuable comments! We sincerely appreciate the time and effort you devoted to reviewing our work. Below, we address each point in detail and hope our responses clarify the contributions and decisions in our paper.
>
> ---
>
> ## Response to W1 part 1/2
>
> To clarify the effectiveness of ILRec that utilizes negatives sampled from intermediate layers instead of the final layer, we conduct a more thorough comparison experiment and more detailed analysis between these two types of negatives.
>
> 1. **Comparison Experiments**
>
> To compare ILRec with the method that directly utilizes negatives from the final layers, we have already provided the comparison results in our paper in Table 2, in which we add $\mathcal{L}_ {CRR}$ and $\mathcal{L}_ {CPD}$ when utilizing final layer's negatives to ensure a fair comparison with ILRec.
>
> Following your suggestion, we also conduct expriments that omit distillation and collaborative loss, only keep $\mathcal{L}_ {CPO}$ to penalize negative tokens for these two methods. All the results are listed in the following table. In the table, **FL** denotes sampling negatives from the **F**inal **L**ayer, and **IL** denotes sampling negatives from the **I**ntermediate **L**ayers. We also indicate the training loss applied for each configuration alongside FL and IL.
>
> **Table 1: Performance comparison between Final layer negatives and Intermediate layers negatives on Amazon Instrument.**
>
> | Method                              | BIGRec |        | LCRec  |        |
> | ----------------------------------- | ------ | ------ | ------ | ------ |
> |                                     | H@10    | N@10    | H@10    | N@10    |
> | SFT                                 | 0.1004 | 0.0799 | 0.1062 | 0.0832 |
> | FL + $\mathcal{L}_ {CPO}$                     | 0.1029 | 0.0814 | 0.1097 | 0.0844 |
> | IL + $\mathcal{L}_ {CPO}$                | 0.1048 | 0.0827 | 0.1118 | 0.0863 |
> | FL + $\mathcal{L}_ {CPO}$ + $\mathcal{L}_ {CPD}$ + $\mathcal{L}_ {CRR}$ | 0.1059 | 0.0839 | 0.1115 | 0.0866 |
> | IL + $\mathcal{L}_ {CPO}$ + $\mathcal{L}_ {CPD}$ + $\mathcal{L}_ {CRR}$                              | 0.1091 | 0.0856 | 0.1143 | 0.0889 |
>
> The results above demonstrate that **utilizing negatives from intermediate layers will achieve better performance and effectiveness rather than final layer's negatives.**
> Below we will give a detailed analysis and comparisons between negatives extracted from the final layer and from intermediate layers.

---

> ### Author Response · Authors · 2025-11-27
> **Rebuttal by Authors**
>
> ## Response to W1 part 2/2
>
> 2. **Detailed Analysis**
>
> To further clarify and analyze why using negatives from the final layer underperforms compared to intermediate-layer negatives in our experiments, we conduct additional studies and summarize our findings as follows:
>
> - **Negatives from the final layer are more likely to be false negatives or overly hard negatives**, which can interfere with the model’s learning process.
>
> - **The SFT loss already penalizes hard negatives at the final layer**, so applying additional penalties to these negatives provides only marginal improvements.
>
> (1) The final layer exhibits substantially stronger recommendation capability than intermediate layers. Consequently, high-probability negatives extracted from the final layer often correspond to items that the model implicitly considers potentially relevant to the user. In other words, these samples are more prone to being false negatives. Aggressively penalizing such items can distort the learned user preference representation and lead to degraded performance.
>
> To further validate that **negatives from the final layer are more likely to be false negatives compared with those from intermediate layers**, we employed two external models—a semantic matching model (E4SRec[1]) and a collaborative filtering model (SASRec)—to measure the similarity between sampled items and their corresponding users from semantic and collaborative perspectives.
>
> Specifically, we randomly selected 3 sets of test samples in the Instrument Dataset, each with 50 test data samples. Then, we beam search intermediate layers and the final layer to generate their top-5 non-positive predicted items over each test data, respectively. After sampling, we computed user–item similarity scores of these items using both semantic and CF models and averaged the results over three test sample sets.
>
> **Table 2: User-Item similarity comparison between Final layer negatives and Intermediate layers negatives.**
>
> |                          | Final Layer | Intermediate Layers |
> | ------------------------ | ----------- | ------------------- |
> | Semantic Model（E4SRec） | **0.8232**      | 0.7814              |
> | CF Model（SASRec）       | **0.7559**      | 0.7380              |
>
> Across both semantic and collaborative similarity metrics, **the non-positive predictions from the final layer are more likely to be recommended to the user than those from intermediate layers**. This indicates that **final-layer negatives tend to lie closer to the user’s preference distribution and are therefore more likely to be false negatives**.
>
> （2）Since ILRec directly applies additional penalties to negative tokens during the SFT stage, we analyzed the gradient of LLM’s SFT loss as follows (we set temperature as 1 for simplicity):
>
> $$
>     \nabla_ \theta \mathcal{L}_ {SFT} = - \nabla_ \theta \log \frac{\exp(p(y_t))}{\sum_ {v\in \mathcal{V}} \exp(p(v))}
> $$
>
> $$
>     = - \frac{\sum_ {v' \in \mathcal{V}_ N}\exp (p(v'))}{\sum_ {v \in \mathcal{V}} \exp (p(v))} \nabla_ \theta p(y_t) + \sum_ {v' \in \mathcal{V}_ N}[\frac{\exp (p(v'))}{\sum_ {v \in \mathcal{V}} \exp (p(v))}] \nabla_ \theta p(v'),
> $$
>
> in which $\mathcal{V}_ N$ refers to all the tokens without the ground-truth token $y_t$. In $\mathcal{L}_ {sft}$, **the model already imposes penalties on hard negative tokens at the final layer**. The gradient of each negative token $v'$ is assigned with an extra weighting $\frac{\exp (p(v'))}{\sum_{v \in \mathcal{V}}\exp (p(v))}$, with stronger penalties applied to harder negatives with higher $p(v')$.
>
> Also, as shown in Table 1, **applying extra penalties to these hard negatives in the final-layer loss (**FL** + $\mathcal{L}_{CPO}$ line) yields only marginal improvements compared with the original SFT baseline (**SFT** line)**. In contrast, ILRec outperforms the variant that uses final-layer negatives in both settings—with and without the distillation loss applied to further enhance negative signals—demonstrating the effectiveness and robustness of ILRec’s design.
>
> [1]Li, Xinhang, et al. "E4srec: An elegant effective efficient extensible solution of large language models for sequential recommendation." arXiv preprint arXiv:2312.02443 (2023).

---

> ### Author Response · Authors · 2025-11-27
> **Rebuttal by Authors**
>
> ## Response to W2
>
> Thanks for your suggestions. The gradient of the token-level loss $\mathcal{L}_ {CPO}$ can be formulated as follows:
>
> $$
>     \nabla_ \theta \mathcal{L}_ {CPO} = - \nabla_ \theta \log \frac{\exp(p(y_t))}{\sum_ {v\in \mathcal{V}} \exp(p(v)(1 + \beta w_ v))}
> $$
>
> $$
>     =
>     - \nabla_\theta p(y_t) + \nabla_\theta \log \sum_{v\in \mathcal{V}} \exp(p(v)(1 + \beta w_v))
> $$
>
> $$
>     =  - \nabla_\theta p(y_t) + \frac{\nabla_\theta\sum_{v\in \mathcal{V}} \exp (p(v)(1 + \beta w_v))}{\sum_{v\in \mathcal{V}} \exp (p(v)(1 + \beta w_v))}
> $$
>
> $$
>     = - \frac{\sum_ {v' \in \mathcal{V}_ N}\exp (p(v')(1 + \beta w_ {v'}))}{\sum_ {v \in \mathcal{V}} \exp (p(v)(1 + \beta w_ v))} \nabla_ \theta p(y_t) + \sum_ {v' \in \mathcal{V}_ N}[\frac{(1 + \beta w_ {v'})\exp (p(v')(1 + \beta w_ {v'}))}{\sum_ {v \in \mathcal{V}} \exp (p(v)(1 + \beta w_ v))}] \nabla_ \theta p(v'),
> $$
>
> in which $\mathcal{V}_ N$ refers to all the tokens without the ground-truth token $y_t$. To penalize high probability tokens provided by intermediate layers, ILRec assigns the gradient of each negative token $v'$ an extra weighting $\frac{(1 + \beta w_{v'})\exp (p(v')(1 + \beta w_{v'}))}{\sum_{v \in \mathcal{V}} \exp (p(v)(1 + \beta w_v))}$. For a challenging high probability token $v'$ provided by intermediate layer, the corresponding probability $w_{v'}$ for token $v'$ is also large, leading to larger extra weighting and more decline in the likelihood of generating specific $v'$.

---

> ### Author Response · Authors · 2025-11-27
> **Rebuttal by Authors**
>
> ## Response to W3
>
> Thank you for this insightful comment! We will provide a clearer justification of the CRR module below.
>
> Although the performance gain of CRR may appear modest at first glance, **it is a crucial component of ILRec, as it injects collaborative signals and prevents certain false negatives from being overly penalized**—an issue that becomes particularly important under large-scale negative sampling.
>
> To clearly demonstrate its necessity, we analyze the contribution of CRR from three complementary perspectives:
>
> **(1) Statistical significance test**
>
> To verify that the improvement from the CRR module is robust, we ran each experiment with 15 different random seeds and performed a paired t-test comparing ILRec with and without CRR. Across all metrics, the improvements are **statistically significant (p < 0.01)**, indicating that the gains from CRR are consistent and are not outcomes with random training fluctuations.
>
> **(2) Additional dataset evaluation demonstrates higher gains**
>
> To examine whether CRR’s effectiveness generalizes beyond a single type of dataset, we further evaluate ILRec on additional datasets: Goodreads and Yelp.
>
> **Table 3: Performance of CRR module on Yelp and Goodreads.**
>
> |               | Yelp         |              | Goodreads    |              |
> | ------------- | ------------ | ------------ | ------------ | ------------ |
> |               | H@10          | N@10          | H@10          | N@10          |
> | ILRec         | 0.0445 +4.2% | 0.0323 +3.5% | 0.0681 +1.9% | 0.0537 +1.5% |
> | ILRec w/o CRR | 0.0427       | 0.0312       | 0.0668       | 0.0529       |
>
>
> Results show that the CRR module leads to **larger improvements on Yelp (+4.2% and +3.5% for Hit@10 and NDCG@10)**, and delivers consistent gains across Goodreads. This shows that CRR is effective across multiple datasets, consistently improving performance, and even yielding notably larger gains on certain datasets.
>
> **(3) Case study: how CRR mitigates the penalization of potential false negatives**
>
> We further conduct a case study to investigate how CRR improves ILRec’s performance. Our approach is to **select test samples that require capturing collaborative filtering (CF) signals and compare the performance of ILRec with and without the CRR module on these samples.** This analysis allows us to **evaluate whether CRR effectively incorporates CF information and helps prevent the model from over-penalizing potential false negatives from the perspective of CF models**.
>
> Specifically, we train the following 4 models on the Art dataset using LC-Rec training paradigm at first:
>
> - a CF model (SASRec) $M_{cf}$
> - a SFT model (LC-Rec) $M_{sft}$
> - ILRec without CRR $M_{wocr}$
> - ILRec with CRR $M_{cr}$
>
> Then, we use $M_{cf}$ to select 40 test data samples in the Art dataset, in which $M_{cf}$ achieves Hit@10=1 among these samples
> Finally, we evaluate the remaining three models on these samples. For each model, we report Hit@10 at three checkpoints, selected sequentially according to the model’s training process. The results are summarized below:
>
> **Table 4: Performancce comparison of different models on CF-guided test sets.**
>
> |  Model          | Origin | Ckpt-1 | Ckpt-2 | Ckpt-3 |
> | ---------- | ------ | ------ | ------ | ------ |
> | $M_{sft}$  | 0.05   | 0.3    | 0.4    | 0.425  |
> | $M_{wocr}$   | 0.05   | 0.25   | 0.4    | 0.325  |
> | $M_{cr}$ | 0.05   | 0.425  | 0.525  | 0.525  |
>
>
> These results show that ILRec with CRR consistently captures items favored by CF models, while ILRec without CRR tends to over-penalize these potential false-negative items and even performs worse than the original $M_{sft}$.

---

> ### Author Response · Authors · 2025-11-27
> **Rebuttal by Authors**
>
> ## Response to Q1
>
> ILRec is a general SFT-enhancement framework that can be plugged into various LLM-based recommendation models. It does not modify the original data format, evaluation protocol, or inference pipeline. **Therefore, for full-ranking recommendation, we directly follow the inference and evaluation setups of BIGRec and LC-Rec, both of which restrict generation to the valid item set**.
>
> **1. Inference strategy**
>
> - **BIGRec** uses beam search to generate b item token sequences based on the interaction-history prompt. Then it **encodes the top-1 generated item and all candidate items with the LLM, ranking candidates using the L2 distance between generated item embedding and candidate item embeddings**. The final recommendation list is obtained by selecting the top-k closest items in from candidates, guaranteeing that the recommended item always lies within the candidate set.
>
> - **LC-Rec** imposes explicit constraints on the item code tokens during generation. **By overwriting prefix_allowed_tokens in model.generate, it enforces a constrained beam search where each decoding step is restricted to a valid token set.** This guarantees that every generated token corresponds to a legal item code within the predefined space.
>
> **2. Evaluation pipeline**
>
> - For BIGRec, we use its inference strategy to obtain the top-k candidate items with highest embedding similarity as the generated item list.
>
> - For LC-Rec, we use beam-search to generate b items, each with a generated probabilities. Then we reorder these items using generated probabiliteis and directly extract the top-k items as the generated item list.
>
> - These top-k lists are used to compute the final evaluation metrics Hit@K and NDCG@K.
>
> A full implementation of our inference and evaluation pipeline is available in our supplementary materials.

---

### Official Review · Reviewer_Crkj · 2025-10-30

**Soundness:** 3
**Presentation:** 3
**Contribution:** 3
**Rating:** 6
**Confidence:** 3

**Summary:**

This paper explores how to utilize the intermediate layers of large language models as self-hard negative generators for recommendation. Instead of relying on negative sampling, the method leverages representations from the model’s intermediate layers to automatically identify “almost positive but actually negative” candidates, which serve as informative hard negatives. The approach further integrates cross-layer preference optimization (CPO) and cross-layer distillation (CPD) to better alleviate the intermediate layer information. Extensive experiments on multiple datasets verify the effectiveness of the proposed method.

**Strengths:**

**The motivation for utilizing the intermediate layers as negative generators is reasonable and intuitive.**
This design of mining “almost positive” tokens intuitively captures the informative negative signals without external sampling.

**Comprehensive experiments.**
The effectiveness of the proposed method is validated across several benchmarks, showing consistent gains over existing negative sampling methods and traditional baselines. The ablation results also verify the contribution of each component.

**Weaknesses:**

**Lack of discussion of utilized CF models.**
The design of collaborative reward regularization is quite heuristic and uncontrollable since we can not ensure which CF model for providing rewards is better. Additionally, this paper does not provide a study about the effectiveness of the chosen CF model.

**Lack of analysis on layer selection.**
 Although the method relies on intermediate layers to generate self-hard negatives, the paper does not include experiments or ablations comparing different layer choices. It remains unclear which layers contribute most to performance or how sensitive the method is to this design choice.

**Questions:**

Refer to the weakness part.

---

> ### Author Response · Authors · 2025-11-27
> **Rebuttal by Authors**
>
> Thank you for your positive feedback and insightful comments! We truly appreciate the time you took to review our paper. Below, we address each of your points in detail. We hope these responses help clarify our paper and address your concerns.
>
> ---
>
> ## Response to W1
>
> We thank the reviewer for raising this important point. To address the concern about **choices and effectiveness of CF models**, we have conducted a performance comparison of ILRec using different CF models in Table 4 in Appendix A.5 of our paper. To provide a more comprehensive analysis, here we further extend the evaluation by adding two additional CF models, BPR and Caser, and list key results as follows:
>
> **Table 1: Performance Comparison of ILRec with different CF models on Amazon Instrument.**
>
> |          | CF Model Performance | BIGRec     |            | LCRec      |            |
> | -------- | -------------------- | ---------- | ---------- | ---------- | ---------- |
> |          | H@10                 | H@10       | N@10       | H@10       | N@10       |
> | SASRec   | 0.0698               | **0.1091** | **0.0856** | **0.1143** | **0.0889** |
> | GRU4Rec  | 0.0773               | 0.1085     | 0.0850     | 0.1136     | 0.0878     |
> | BERT4Rec | 0.0681               | 0.1089     | 0.0851     | 0.1142     | 0.0885     |
> | Caser    | 0.0583               | 0.1073     | 0.0842     | 0.1125     | 0.0871     |
> | BPR      | 0.0406               | 0.1068     | 0.0837     | 0.1125     | 0.0867     |
>
>
> These results allow us to clarify that **the choice of CF models is relatively controllable**, and **the final effectiveness of CF models is relatively predictable**, which can be attributed to two observations:
>
> (1) **ILRec is robust to CF models with similar high recommendation quality**.
> As shown in Table 1, CF models with comparable standalone performance (e.g., SASRec, GRU4Rec, BERT4Rec) yield consistently similar improvements when integrated into ILRec. This demonstrates that CRR does not depend on any specific CF model and remains stable improvement across strong CF models.
>
> (2) **Stronger CF models provide stronger rewards and typically yield better final performance**.
> We observe a clear and consistent trend: higher-performing CF models (SASRec, GRU4Rec, BERT4Rec) bring larger gains in ILRec than relatively weaker CF models (Caser, BPR). This indicates that the effectiveness of chosen CF models is not heuristic or unpredictable, but rather correlates with the inherent quality of the CF model.
>
> Following this principle for selecting CF models, ILRec achieves consistent gains across multiple datasets and training paradigms, including both BIGRec and LCRec settings, which demonstrates that **the chosen CF model (SASRec) and the CRR module are both effective and generalizable in various scenarios**.

---

> ### Author Response · Authors · 2025-11-27
> **Rebuttal by Authors**
>
> ## Response to W2:
>
> In ILRec, we use consecutive intermediate layers directly before the final layer as the source for negative generation. In response to your insightful comment, we performed supplementary experiments to more thoroughly examine the effectiveness of this choice by comparing multiple alternative layer choices, as shown below：
>
> **1. Shallow layers vs. Deep layers**
>
> As shown in Figure 2 in our paper, deeper intermediate layers learn more effectively on the recommendation task, while shallower layers produce less reliable predictions. **Negatives from shallow layers are more likely to be noisy, limiting ILRec’s effectiveness**.
>
> To verify this observation, we empirically compared deep consecutive layers (31–29) with medium layers (26-24) and shallow layers (21–19) on Llama-3.1-8B under the LC-Rec paradigm on the Instrument dataset. As shown in Table 2, deep layers outperform medium and shallow counterparts, showing lower cross-entropy loss and higher recommendation performance for ILRec.
>
> **Table 2: Performance comparison with different depth of layers.**
>
> |  Methods                        | H10    | N10    |
> | ------------------------ | ------ | ------ |
> | deep 3 layers     | **0.1091** | **0.0856** |
> | medium 3 layers         | 0.1043 | 0.0820 |
> | shallow 3 layers         | 0.1037 | 0.0813 |
>
> ---
>
> **2. Consecutive layers vs. Non-consecutive / Single layers**
>
> To ensure informative negatives, we select k consecutive layers starting from the final layer. Non-consecutive layers **tend to include weaker layers and lack a principled selection rule**, while using a single layer **reduces stability and limits negative diversity**.
>
> We validate this design by comparing three strategies under the same experimental settings:
> - consecutive layers 31–29
> - three randomly chosen non-consecutive layers from 31–24
> - individual layers 31, 30, and 29.
>
> The results in Table 3 consistently indicate that consecutive deep layers yield the best performance, supporting the robustness and effectiveness of our layer-selection strategy.
>
> **Table 3: Performance comparison with different layer selection strategies on Amazon Instrument.**
>
> |  Methods                        | H10    | N10    |
> | ------------------------ | ------ | ------ |
> | consecutive 3 layers     | 0.1091 | 0.0856 |
> | non-consecutive 3 layers | 0.1084 | 0.0843 |
> | single $L_ {31}$          | 0.1067 | 0.0839 |
> | single $L_ {30}$          | 0.1080 | 0.0844 |
> | single $L_ {29}$          | 0.1075 | 0.0836 |
>
>
> **3. The guidance for selecting appropriate k**
>
> We additionally examine how ILRec’s performance varies with different values of k, providing insights for choosing an appropriate setting in practice.
>
> **Table 4: Performance comparison with different k among different model backbones on Amazon Instrument.**
>
> | k           | 1             | 2                 | 3                 | 4             | 5             | 6             | 7      |
> | ----------- | ------------- | ----------------- | ----------------- | ------------- | ------------- | ------------- | ------ |
> | Llama3.2-1B | _0.0972_ | _**0.0976**_ | _0.0968_     | 0.0950        | 0.0953        | 0.0938        | 0.0947 |
> | Llama3.2-3B | 0.0993        | _0.1019_     | _**0.1024**_ | _0.1022_ | 0.1008        | 0.1013        | 0.1002 |
> | Llama3.1-8B | 0.1067        | _0.1081_     | **_0.1091_** | _0.1084_ | _0.1088_ | _0.1075_ | 0.1057 |
>
>
> From Table 4, we can summarize the guidance for selecting k as follows:
> - **The optimal k consistently lies within intermediate layers closed to the final layer(<= $5^{th}$ layer)**, as shown in the bold data in Table 4. These deep layers have sufficient predictive capability to provide informative negatives, compared with shallower layers.
>
> - **ILRec is robust to a range of nearby k values**, as shown in the italic data in Table 4. Performance around the optimal k varies smoothly, and neighboring values also yield comparable gains, meaning that only a small range of k needs to be tested to achieve stable performance improvements.
>
> - **Larger models support a broader effective range of k**.
> Intermediate layers in larger models generate stronger predictions and higher-quality negative signals. Therefore, while 1B models have a narrow effective range of k for achieving stable performance improvement, 3B and 8B models benefit from a broader selection of k.

---

### Official Review · Reviewer_cg7x · 2025-11-01

**Soundness:** 3
**Presentation:** 4
**Contribution:** 3
**Rating:** 8
**Confidence:** 4

**Summary:**

This paper addresses the limitations of indistinctiveness and low informativeness of negative samples in existing LLM-based recommendation methods. It proposes ILRec, a novel preference fine-tuning framework that extracts fine-grained self-hard negative signals from the intermediate layers of LLMs. The framework consists of three core components: 1) Self-hard negative extraction from intermediate layers, which selects high-probability non-ground-truth tokens as token-level negative signals; 2) Cross-layer preference fine-tuning, including cross-layer preference optimization (integrating negative signals into cross-entropy loss with penalty coefficients) and cross-layer preference distillation (using the final layer to supervise intermediate layers); 3) Collaborative reward regularization, which employs a lightweight collaborative filtering (CF) model to assign token-level rewards and avoid over-penalization of false negatives. Extensive experiments on three Amazon datasets (Musical Instruments, Arts, Crafts and Sewing, Video Games) demonstrate that ILRec outperforms traditional sequential recommendation models and LLM-based baselines (e.g., BIGRec, LC-Rec, SDPO) on metrics like Hit@k and NDCG@k. Ablation studies verify the effectiveness of each component, and further analysis confirms its generalizability across different model backbones and recommendation tasks.

**Strengths:**

## 1. Novelty.
- The idea of using LLM intermediate layers as dynamic self-hard negative generators is innovative. It breaks through the limitation of offline static negative sampling in existing DPO-based methods and provides fine-grained token-level negative signals, which is a creative combination of LLM internal structure characteristics and recommendation task requirements.
- The cross-layer preference optimization and distillation mechanism realizes the mutual promotion of negative signal quality and model learning effect, and the introduction of collaborative reward regularization effectively solves the problem of false negative over-penalization, forming a complete and self-consistent technical framework.

## 2. Clarity.
- The paper structure is logical: it starts with the limitations of existing methods, introduces the core idea, elaborates on the technical details of each component, and presents experimental results and analysis in turn. The logical chain from problem to solution to verification is complete.
- The figures and tables are intuitive, which helps readers understand the core design and experimental results. The mathematical formulas are clearly defined and the symbols are consistent throughout the paper.

## 3. Significance.
- For the field of LLM-based recommendation, it provides a new effective path for negative sample construction, which can effectively handle the challenge of large negative item spaces and improve the model's ability to capture fine-grained user preferences.

**Weaknesses:**

## 1. Insufficient analysis of intermediate layer selection
- The optimal number of intermediate layers (k) is determined through experiments, but there is no theoretical guidance for the selection range of k. For different scales of LLMs (e.g., 1B vs. 3B vs. 8B), the suitable k may vary significantly, and the paper does not provide relevant analysis.

- The paper selects "consecutive intermediate layers before the final output layer" as candidate layers but does not explain why consecutive layers are preferred over non-consecutive ones. It also lacks analysis on how the position of intermediate layers (e.g., shallow vs. deep) affects the quality of negative signals.

## 2. Lack of analysis on computational cost details.
- Although the paper mentions that ILRec does not introduce excessive training time costs, it does not provide detailed computational overhead analysis of key components.

## 3. Insufficient exploration of application boundaries.
- The experiment of this study is limited to the Amazon review dataset and lacks data from other scenarios such as Goodreads and Yelp.

**Questions:**

## 1. Technical Questions

- For collaborative reward regularization: If the CF model's recommendation results are seriously inconsistent with the LLM's preference, how will it affect ILRec's performance? Have you designed corresponding adaptive mechanisms?

## 2. Experimental Suggestions

- Add experiments on LLMs of different scales to verify the generalizability of ILRec across model sizes and analyze the optimal k value for different models.

- Provide detailed computational cost metrics and compare them with baselines to clarify the efficiency advantages of ILRec in practical applications.

**Details Of Ethics Concerns:**

No concern

---

> ### Author Response · Authors · 2025-11-27
> **Rebuttal by Authors**
>
> We sincerely thank you for the valuable time and effort dedicated to reviewing our paper. We appreciate your insightful comments and constructive feedback.
> In response to your questions, we have carefully addressed them below.
>
> ---
>
> ## Response to W1 part 1/2
>
> To clarify and analyze our intermediate-layer selection strategy, as well as its applicability across models of different sizes, we organize our response into the following points:
>
> **1. The optimal range of k for different models and the guidance for selecting k**
>
> Regarding your concern about the optimal range of k for different models and the theoretical guidance for selecting k, we conducted an additional analysis systematically examine the sensitivity of ILRec to the value of k.
>
> Specifically, while keeping all other hyperparameters fixed, we varied k from 1 to 7 and evaluated different models' performance with the ILRec task on the Instrument dataset, using Hit@10 as the main metric. The results are summarized in Table 1.
>
> **Table 1: Performance comparison with different k among different model backbones on Amazon Instrument.**
> | k           | 1             | 2                 | 3                 | 4             | 5             | 6             | 7      |
> | ----------- | ------------- | ----------------- | ----------------- | ------------- | ------------- | ------------- | ------ |
> | Llama3.2-1B | _0.0972_ | _**0.0976**_ | _0.0968_     | 0.0950        | 0.0953        | 0.0938        | 0.0947 |
> | Llama3.2-3B | 0.0993        | _0.1019_     | _**0.1024**_ | _0.1022_ | 0.1008        | 0.1013        | 0.1002 |
> | Llama3.1-8B | 0.1067        | _0.1081_     | **_0.1091_** | _0.1084_ | _0.1088_ | _0.1075_ | 0.1057 |
>
>
> From Table 1, we can summarize the guidance for selecting k as follows:
> - **The optimal k consistently lies within intermediate layers closed to the final layer(<= $5^{th}$ layer)**, as shown in the bold data in Table 1. These deep layers have sufficient predictive capability to provide informative negatives, compared with shallower layers.
>
> - **ILRec is robust to a range of nearby k values**, as shown in the italic data in Table 1. Performance around the optimal k varies smoothly, and neighboring values also yield comparable gains, meaning that only a small range of k needs to be tested to achieve stable performance improvements.
>
> - **Larger models support a broader effective range of k**.
> Intermediate layers in larger models generate stronger predictions and higher-quality negative signals. Therefore, while 1B models have a narrow effective range of k for achieving stable performance improvement, 3B and 8B models benefit from a broader selection of k.
>
> The detailed performance of ILRec across the three models further demonstrates its generalizability as follows:
>
> **Table 2: Performance of ILRec on different scales of models on Amazon Instrument.**
>
> |    Model               | BIGRec |      | LCRec |      |
> | ----------------- | ------ | ---- | ----- | ---- |
> |                   | H@10     | N@10  | H@10    | N@10  |
> | Llama3.2-1B SFT   | 0.0942 | 0.0731 | 0.1007 | 0.0670 |
> | Llama3.2-1B ILRec | 0.0976 | 0.0772 | 0.1049 | 0.0721 |
> | Llama3.2-3B SFT   | 0.0968 | 0.0748 | 0.1021 | 0.0726 |
> | Llama3.2-3B ILRec | 0.1024 | 0.0791 | 0.1083 | 0.0782 |
> | Llama3.1-8B SFT   | 0.1004 | 0.0799 | 0.1062 | 0.0832 |
> | Llama3.1-8B ILRec | 0.1091 | 0.0856 | 0.1143 | 0.0889 |

---

> ### Author Response · Authors · 2025-11-27
> **Rebuttal by Authors**
>
> ## Response to W1 part 2/2
>
> **2. The rationale of layer selection strategies**
>
> To further justify our design choices for selecting consecutive intermediate layers right before the final layer for negative sampling, we conduct two sets of experiments as follows:
>
> **(1) Consecutive layers vs. Non-consecutive / Single layers**
>
> To ensure informative negatives, we select k consecutive layers starting from the final layer as negative generator in ILRec. Non-consecutive layers **tend to include weaker layers and lack a principled selection rule**, while using a single layer **reduces stability and limits diversity of negatives**.
>
> We validate this design by comparing three strategies under the same experimental settings:
> - consecutive layers 31–29
> - three randomly chosen non-consecutive layers from 31–24
> - individual layers 31, 30, and 29.
>
> The results in Table 3 consistently indicate that consecutive deep layers yield the best performance, supporting the robustness and effectiveness of our layer-selection strategy.
>
> **Table 3: Performance comparison with different layer selection strategies on Amazon Instrument.**
>
> |  Methods                        | H@10    | N@10    |
> | ------------------------ | ------ | ------ |
> | consecutive 3 layers     | **0.1091** | **0.0856** |
> | non-consecutive 3 layers | 0.1084 | 0.0843 |
> | single $L_ {31}$          | 0.1067 | 0.0839 |
> | single $L_ {30}$          | 0.1080 | 0.0844 |
> | single $L_ {29}$          | 0.1075 | 0.0836 |
>
> **(2) Shallow layers vs. Deep layers**
>
> As shown in Figure 2 in our paper, deeper intermediate layers learn more effectively on the recommendation task, while shallower layers produce less reliable predictions. **Negatives from shallow layers are more likely to be noisy, limiting ILRec’s effectiveness**.
>
> To verify this observation, we empirically compared deep consecutive layers (31–29) with medium layers (26-24) and shallow layers (21–19) on Llama-3.1-8B under the LC-Rec paradigm on the Instrument dataset. As shown in Table 4, deep layers outperform medium and shallow counterparts, showing lower cross-entropy loss and higher recommendation performance for ILRec:
>
> **Table 4: Performance comparison with different depth of layers.**
>
> |  Methods                        | H@10    | N@10    |
> | ------------------------ | ------ | ------ |
> | deep 3 layers     | **0.1091** | **0.0856** |
> | medium 3 layers         | 0.1043 | 0.0820 |
> | shallow 3 layers         | 0.1037 | 0.0813 |
>
> ---
>
> ## Response to W2
>
> For computational cost analysis, we provide a detailed comparison between ILRec and several DPO-based recommendation baselines, including S-DPO, RosePO, and SPRec. The key results for both the Instrument and Art datasets are summarized below, as well as in Table 5 of Appendix A.6 in our paper.
>
> **Table 5: Efficiency analysis of different methods.**
>
> | Model | Instrument       |     |        | Art |        |       |
> | ------------------------ | ------ | ------ | ------------------------ | ------ | ------ |------ |
> | |Time       | Epoch     | Sample       | Time | Epoch       | Sample      |
> | S-DPO   | 7.25 h | 8(5 + 3)   | 1 | 10.25 h   | 8(5 + 3)  | 1   |
> | RosePO | 6.6 h | 8(5 + 3)   | 1 | 9.53 h   | 8(5 + 3) | 1   |
> | SPRec  | 7.8 h | 11(5 + 3 * 2)   | 3 | 11.5 h   | 11(5 + 3 * 2) | 3 |
> | ILRec  | 4.2 h | 5   | 0 | 7.46 h   | 5 | 0 |
>
> To ensure a fair comparison, we evaluated:
> - Time: total training time
> - Epoch:  number of epochs (including SFT + DPO * iterations)
> - Sample: number of negative-sample generation steps required by each method
>
> ILRec achieves the highest training efficiency among all compared preference-optimization methods. This efficiency stems from its ability to **directly extract informative negative signals from its own intermediate layers within the SFT stage**, eliminating both (i) the extra preference-alignment phase and (ii) repeated negative-sample generation. As a result, ILRec substantially reduces overall training time while maintaining strong recommendation performance.

---

> ### Author Response · Authors · 2025-11-27
> **Rebuttal by Authors**
>
> ## Response to W3
>
> To address your concern regarding dataset diversity, we additionally evaluate our method on two more benchmark datasets, Goodreads and Yelp, which represent different domains and interaction patterns. The results of BIGRec training paradigm on these two datasets are reported as follows:
>
> **Table 6: Performance of ILRec on Yelp and GoodReads.**
>
> |        | Yelp   |        |        |        | GoodReads |        |        |        |
> | ------ | ------ | ------ | ------ | ------ | --------- | ------ | ------ | ------ |
> |        | H@5    | N@5    | H@10   | N@10   | H@5       | N@5    | H@10   | N@10   |
> | BIGRec | 0.0247 | 0.0163 | 0.0383 | 0.0279 | 0.0350    | 0.0248 | 0.0619 | 0.0502 |
> | ILRec  | 0.0288 | 0.0191 | 0.0445 | 0.0323 | 0.0378    | 0.0294 | 0.0681 | 0.0537 |
>
>
> As shown in Table 6, our method consistently outperforms the SFT baseline across these datasets. In particular, on the Yelp dataset, ILRec achieves the largest improvement on NDCG@5, outperforming BIGRec by 17.1%, demonstrating its robustness and generalizability beyond the Amazon review datasets.
>
> ---
>
> ## Response to Q1
>
> Thank you for highlighting this perspective. In ILRec, the CF model in the CRR module serves to prevent high-CF-score items—i.e., potential false negatives—from being overly penalized. Large discrepancies between CF and LLM predictions, as you mentioned, may arise for two reasons.
>
> (1) **CF model is too weak or misaligned with user preferences**, which fail to capture false negatives and provided limited or noisy CF signals for LLMs to learn. We mitigate this by **reducing CRR loss weight** or **using a stronger CF models**.
>
> (2) **The LLM has not yet captured CF signals**, in which the CF model provides complementary information that may improve the ILRec’s recommendation ability.
> In this case, we **slightly increase the CRR loss weight** to strengthen the incorporation of complementary CF signals.
>
> We appreciate the reviewer for raising this insightful question. Understanding and adapting to CF–LLM discrepancies is indeed an important direction. In future work, we plan to investigate:
> - **Adaptive CRR weighting**, where the influence of CF signals is dynamically adjusted based on the real-time alignment between CF predictions and the LLM’s token-level probabilities;
> - **Confidence-aware CF filtering**, so that only high-confidence CF predictions contribute to the regularization term.
>
> ---
>
> ## Response to Q2
> Please refer to responses for W1 and W2.
>
> ---

---

### Author Response · Authors · 2025-12-03
**Summary of Our Responses and Revisions**

Dear Area Chairs,

We sincerely appreciate your time and effort in managing our submission, and we are grateful to the reviewers for their insightful and constructive comments. In response to their main concerns, we have conducted extensive additional experiments and revised the manuscript. We believe these new results and updates will **effectively address the reviewers’ key points and further strengthen the contribution of our work**. Below is a summary of our key clarifications, new experimental evidence, and major revisions made during the rebuttal.

---

### **1. Our Strengths**
We are truly encouraged that the reviewers acknowledged the contributions and strengths of our work, which we briefly summarize below:

- **Novelty of ILRec**: All reviewers recognized the novelty of extracting negatives from intermediate layers for LLM-based recommendation. Reviewer cg7x highlighted that ILRec overcomes the limitations of offline static negative sampling in existing DPO-based methods and creatively leverages LLM internal structures for recommendation tasks. Reviewers Crkj, t9VT, and UTWL further agreed that using intermediate layers as negative generators is well-motivated, reasonable, and intuitive.

- **Effectiveness of Methodology**: Reviewer cg7x acknowledged the effectiveness of method for utilizing extracted negatives, with the cross-layer preference fine-tuning mechanism and collaborative reward regularization module forming a complete and self-consistent technical framework.

- **Generality of ILRec and Comprehensive Experiments**: Reviewers Crkj and t9VT further recognized the strong generality of ILRec across different training paradigm(BIGRec + LC-Rec), backbone models, and collaborative models, as well as the completeness and rigor of our experimental evaluation on several benchmark.


- **Clarity and Presentation**: Reviewers cg7x and UTWL found our paper well-structured and easy to follow. The logical chain from problem, solutions to verification is complete.

---

> ### Author Response · Authors · 2025-12-03
> **Summary of Our Responses and Revisions (Continued)**
>
> ### 2. Responses to Concerns
> Here we summarize our responses and resolutions to the key concerns raised by the reviewers.
>
> **2.1. The Effectiveness of CF Models(Addressing Reviewers Crkj & t9VT & UTWL)**: The key concern was whether the Collaborative Reward Regularization(CRR) module is important for ILRec, and how to effectively choose CF models for CRR module.
>
> - (1) We extended our experiment in Appendix A.5 in our paper to **compare the performances of ILRec with various CF models**   (SASRec, BERT4Rec, GRU4Rec, Caser and BPR). The results indicate that ILRec is robust to CF models with consistently strong recommendation quality **(demonstrating the effectiveness of CF models)**, and stronger CF models yield better final performance for ILRec **(providing practical guidance for selecting CF models)**.
>
>
> - (2) We conducted **statistical significance tests** and **evaluated CRR on two other datasets** (Goodreads and Yelp). With results showing p < 0.01 and a 4.2% HIT@10 improvement on Yelp, the effectiveness of the CRR module is demonstrated to be both consistent and statistically significant.
>
> - (3) We further conducted **a case study to evaluate ILRec's performance on test samples that require CF information (selected by SASRec)**. ILRec with CRR module outperforms other counterparts, proving its effectiveness.
>
>
> **2.2. Extracting negatives from final vs. intermediate layers(Addressing Reviewers t9VT & UTWL)**: The primary concern was whether extracting negatives from intermediate layers is better than from the final layer.
>
> - (1) We extended our ablation study by **comparing ILRec using final-layer negatives versus intermediate-layer negatives, both without distillation or collaborative loss**. The results show that intermediate-layer negatives consistently outperform final-layer negatives.
>
> - (2) We further conducted detailed analysis to explain this phenomenon.
>   - First, **we examined the intrinsic differences between final-layer and intermediate-layer negatives**. By evaluating their similarity to user embeddings using external semantic and CF models, we found that **negatives from the final layer are more likely to be false negatives**.
>   - Second, **we analyzed the model’s training dynamics**. Through gradient analysis of the SFT loss, we observed that **SFT already imposes a certain level of penalty on hard final-layer negatives during training. Further penalization yields only marginal improvements**.
>
>
> **2.3. Strategies of Layer Selection(Addressing Reviewers cg7x and Crkj)**: The key concern centered on why we select consecutive intermediate layers right before the final layer for negative extraction, rather than alternative strategies, as well as how to choose an appropriate k for models of different scales.
>
> - (1) We compared our layer selection strategy with **multiple alternatives—including selecting non-consecutive layers, single-layer, and shallow layers**. The results confirm the effectiveness of our proposed approach.
>
> - (2) We conducted experiment to **find the optimal range of k for different models and the guidance for selecting k**. We varied k from 1 to 7 on three scales of models (1B, 3B & 8B). We found that:
>   - The optimal k consistently lies within intermediate layers closed to the final layer.
>   - ILRec is robust to a range of nearby k values.
>   - Larger models support a broader effective range of k.
>
>
> **2.4. Other Concerns**:
> In addition to the main issues discussed above, we also thoroughly addressed the remaining concerns raised by each reviewer.
>
> - For reviewer cg7x, we provided detailed computational cost analysis, and applied ILRec to other datasets.
>
> - For reviewer Crkj, we provided our reponses of the main concerns in 2.1 and 2.3.
>
> - For reviewer t9VT, we conducted a gradient-based analysis of the CPO loss, and clarified our evaluation methodology with further details.
>
> - For reviewer UTWL, we introduced RLVR-based methods as additional baselines, and demonstrated that token-level DPO offers more precise reward signals than sequence-level rewards. We also conducted hyperparameter sensitivity analyses for $\alpha$, explored alternative logit-merging methods, and evaluated ILRec with CF-generated negatives.

---

> ### Author Response · Authors · 2025-12-03
> **Summary of Our Responses and Revisions (Continued)**
>
> ### 3.Revisions in the Manuscript
>
> We have also uploaded a revised version of the manuscript. The primary updates focus on expanding the experimental evaluation to address reviewer concerns. All revisions are highlighted in blue. The major modifications are as follows:
>
> - Additional ablation experiments and discussions comparing negatives extracted from the final layer versus intermediate layers, as presented in Table 2 and Section 3.3.
> - Experiments and discussions examining the effectiveness of different CF models for ILRec and demonstrating the robustness of our approach across various CF models, as detailed in Appendix A.5.
> - Gradient analysis of the cross-layer preference optimization loss, provided in Appendix A.8.
> - Experiments and discussions on alternative layer-selection strategies, along with practical guidance for choosing $k$ across different model scales, as detailed in Appendix A.9.
>
> ---
>
> We believe that we have comprehensively addressed all distinct concerns raised by the reviewers and made substantial updates to the manuscript.
>
> We hope this summary clarifies the review process and assists in your decision. Thank you once again for your valuable time and consideration!
>
> Best regards,
>
> Authors of Submission 18825

---

### Meta-Review · Area_Chair_bjfg · 2026-01-02

**Summary:**

Reviewers found the idea of extracting token-level hard negatives from intermediate layers interesting, but several concerns remained central to their assessments and support a reject recommendation. A key issue was whether intermediate-layer negatives are truly necessary or meaningfully different from simpler alternatives such as using final-layer high-probability tokens or self-generated negatives, and reviewers requested clearer ablations and deeper analysis to justify the design. Reviewers also questioned the heuristic and added complexity of the collaborative reward regularization module, noting that its gains appeared modest and that performance could depend on the choice and alignment of the external CF model. In addition, the empirical evidence was viewed as incomplete: while the paper reports main results on multiple datasets, key ablations and analyses are presented only on a subset of the benchmark, making it harder to assess whether the claimed mechanisms hold consistently. Additional concerns included limited initial justification for layer-selection and hyperparameter choices, incomplete comparison against more recent RLVR-style LLM recommendation methods, and insufficient clarity on practical evaluation and efficiency details.

**Reviewer Concerns:**

The rebuttal does address many concrete reviewer requests. It adds ablations on layer choice and k (deep vs shallow, consecutive vs non-consecutive/single layers, model-scale sensitivity), clearer intermediate- vs final-layer negative comparisons (including controls without distillation/CRR), cost reporting versus DPO-style baselines, and broader evaluation beyond Amazon (Yelp/Goodreads). It also strengthens the CRR evidence with CF-model sweeps, significance tests, and a targeted case study, and clarifies inference/evaluation details, α sensitivity, and alternative layer-logit aggregation schemes, with at least one RLVR-style baseline added.

However, the core issues motivating rejection remain. The main contribution still feels like an interesting empirical effect without a convincing underlying principle: it is not fully clear why intermediate-layer negatives are fundamentally necessary or qualitatively better than well-designed final-layer/self-generated token-level negatives. The full method is also relatively complex (CPO+CPD+CRR, multiple knobs, reliance on an external CF model) for incremental gains, and baseline coverage/positioning against the broader frontier (RLVR-style and stronger token-level preference baselines) is still not decisively settled.

**Reviewer Scores:**

- cg7x (8 → 8): Their main asks (layer/k analysis, cost, broader datasets) were addressed, so they would likely keep an accept score rather than increase it further.

 - Crkj (6 → 6): The rebuttal mitigates their two stated weaknesses (CF choice and layer selection), but they may still view the overall design as somewhat heuristic, so they likely stay at a mild accept.

 - t9vT (4 → 4 or 5): The added controls and loss/CRR analyses reduce several objections, but their core skepticism about the method design and necessity of intermediate-layer negatives may persist, so at most a small upward move.

 - UTWL (4 → 4 or 5): The rebuttal addresses many items (ablations, sensitivity, partial RLVR comparison), but broader baseline completeness and conceptual distinction concerns can remain, so at most a modest increase.

---

### Decision · Program_Chairs · 2026-01-26

Reject